# The effect of viscosity and diffusion on the HO₂ uptake by sucrose and secondary organic aerosol particles

Pascale S. J. Lakey[1,2], Thomas Berkemeier[2], Manuel Krapf[3], Josef Dommen[3], Sarah S. Steimer[3], Lisa K. Whalley[1,4], Trevor Ingham[1,4], Maria T. Baeza-Romero[5], Ulrich Pöschl[2], Manabu Shiraiwa[2,6], Markus Ammann[3] and Dwayne E. Heard[1,4, *]

*[1] School of Chemistry, University of Leeds, Woodhouse Lane, Leeds, LS2 9JT, UK*

*[2] Multiphase Chemistry Department, Max-Planck-Institute for Chemistry, Hahn-Meitner-Weg 1, 55128 Mainz, Germany.*

*[3] Paul Scherrer Institute, Villigen, Switzerland.*

*[4] National Centre for Atmospheric Chemistry, University of Leeds, Woodhouse Lane, Leeds, LS2 9JT, UK*

*[5]Escuela de Ingeniería Industrial de Toledo, Universidad de Castilla la Mancha, Avenida Carlos III s/n Real Fábrica de Armas, Toledo, 45071, Spain*

*[6]Department of Chemistry, University of California Irvine, CA 92617, United States*

*\* Corresponding author: Dwayne Heard (D.E.Heard@leeds.ac.uk)*

## Abstract

We report the first measurements of HO₂ uptake coefficients, $\gamma$, for secondary organic aerosol particles (SOA) and for the well-studied model compound sucrose which we doped with copper (II). Above 65% relative humidity (RH), $\gamma$ for copper (II) doped sucrose aerosol particles equalled the surface mass accommodation coefficient $\alpha = 0.22 \pm 0.06$ but decreased to $\gamma = 0.012 \pm 0.007$ upon decreasing the RH to 17 %. The trend of $\gamma$ with RH can be explained by an increase in aerosol viscosity and the contribution of a surface reaction, as demonstrated using the kinetic multi-layer model of aerosol surface and bulk chemistry (KM-SUB). At high RH the total uptake was driven by reaction in the near-surface bulk limited by mass accommodation whilst at low RH it was limited by surface reaction. SOA from two different precursors, α-pinene and 1,3,5- trimethylbenzene (TMB), was investigated, yielding low uptake coefficients of $\gamma < 0.001$ and $\gamma = 0.004 \pm 0.002$, respectively. It is postulated that the larger values measured for TMB derived SOA compared to α-pinene derived SOA are either due to differing

viscosity, a different liquid water content of the aerosol particles or a $HO_2$ + $RO_2$ reaction occurring
within the aerosol particles.

## Introduction


OH and $HO_2$ radicals play a vital role in atmospheric chemistry by controlling the oxidative
capacity of the troposphere, with $HO_2$ acting as a short-lived reservoir for OH. Oxidation by the OH
radical determines the lifetime and concentrations of many trace gases within the troposphere such as
$NO_x$ (NO and $NO_2$), $CH_4$ and volatile organic compounds (VOCs). The reaction of $HO_2$ with NO also
constitutes an important source of ozone, which is damaging to plants, a respiratory irritant and a
greenhouse gas (Pöschl and Shiraiwa, 2015;Fowler et al., 2009). It is therefore important to have a
thorough understanding of the reactions and processes that affect $HO_x$ concentrations. However, during
field campaigns $HO_2$ concentrations have sometimes been measured as being lower than the
concentrations predicted by constrained box models implying a missing $HO_2$ sink, which has often been
attributed to $HO_2$ uptake by aerosol particles (e.g. (Kanaya et al., 2007;Mao et al., 2010;Whalley et al.,
46    2010)).

SOA is generated from low-volatility products formed by the oxidation of VOCs, and it
accounts for a large fraction of the organic matter in the troposphere. For example, in urban areas it can
account for up to 90 % of the organic particulate mass (Kanakidou et al., 2005;Lim and Turpin, 2002).
Lakey et al. (2015a) previously measured the $HO_2$ uptake coefficient onto single component organic
aerosol particles as ranging from $\gamma < 0.004$ to $\gamma = 0.008 \pm 0.004$ unless elevated transition metal ions,
that catalyse the destruction of $HO_2$, were present within the aerosol. Taketani et al. (2013) and Taketani
and Kanaya (2010) also measured the $HO_2$ uptake coefficient onto dicarboxylic acids ($\gamma = 0.02 \pm 0.01$
to $\gamma = 0.18 \pm 0.07$) and levoglucosan ($\gamma < 0.01$ to $\gamma = 0.13 \pm 0.03$) over a range of humidities. However,
there are currently no measurements of the $HO_2$ uptake coefficient onto SOA published in the literature.
Using the kinetic multi-layer model of aerosol surface and bulk chemistry (KM-SUB), Shiraiwa
et al. (2011b) have shown that the bulk diffusion of a species within an aerosol matrix can have a large
impact on a measured uptake coefficient. Diffusion coefficients of a particular species within a particle
are related to the viscosity of that particle with larger diffusion coefficients in less viscous particles.
Traditionally, the relationship between viscosity and diffusion coefficients is given by the Stokes-
Einstein equation, although this relation was found to break down for concentrated solutions and
solutions near their glass transition temperature or humidity (Champion et al., 1997;Power et al., 2013).
Zhou et al. (2013) have also shown that the rate of heterogeneous reaction of particle-borne
benzo[a]pyrene (BaP) with ozone within SOA particles was strongly dependent upon the bulk
diffusivity of the SOA. Along the same lines, Steimer et al. (2015) and Steimer et al. (2014)
demonstrated a clear link between the ozonolysis rates of shikimic acid and the changing diffusivity in
the transition between liquid and glassy states. Previous measurements of both $N_2O_5$ uptake coefficients
and $HO_2$ uptake coefficients onto humic acid aerosol particles and $N_2O_5$ uptake coefficients onto
malonic acid and citric acid aerosol particles have shown much lower uptake coefficients at low relative
humidities compared to higher humidities (Badger et al., 2006;Thornton et al., 2003;Lakey et al.,
2015a;Gržinić et al., 2015). However, viscosity effects have not been investigated systematically for
$HO_2$ uptake, and the first aim of this paper was to investigate whether a change in aerosol viscosity,
exemplified using the well-studied model compound sucrose (Berkemeier et al., 2014;Price et al.,
2014;Zobrist et al., 2011), could impact the $HO_2$ uptake coefficient. The second aim of this study was
to measure the $HO_2$ uptake coefficient onto two different types of SOA representative of biogenic and
anthropogenic SOA. α-pinene is the major terpene that forms biogenic SOA, while 1,3,5-
trimethylbenzene (TMB) is representative of alkyl benzenes which are the most abundant aromatic
hydrocarbons and form anthropogenic SOA (Calvert et al., 2002;Qi et al., 2012). SOA is known to be
highly viscous with viscosities of $10^3 - 10^6$ Pa s at 50 % RH (Renbaum-Wolff et al., 2013).

## Experimental


The general experimental setup for the Leeds aerosol flow tube and the data analysis
methodology to determine values of γ have previously been discussed in detail by George et al. (2013).
This is the same experimental setup and data analysis methodology that was used for the copper (II)
doped sucrose experiments, which were also performed at the University of Leeds. Therefore, only a
brief description of the setup is included below, with the emphasis being on changes made to the
apparatus for the SOA experiments undertaken at the Paul Scherrer Institute (PSI), for which a
schematic is shown in Figure 1. For all experiments the $HO_2$ radical was released at the end of an
injector which was moved backwards and forwards along an aerosol flow tube. The flow from the
injector was $1.32 \pm 0.05$ slpm. For the copper doped sucrose experiments the humid aerosol flow was
$1.0 \pm 0.1$ slpm, and was mixed with a much drier flow (with the humidity of this flow being controlled
by mixing a flow from a water bubbler with a dry flow in different ratios) of $3.0 \pm 0.3$ slpm within a
conditioning flow tube for approximately ten seconds before entering the aerosol flow tube. Nitrogen
was used for all of these flows. For the SOA experiments the flow from the smog chamber or Potential
Aerosol Mass (PAM) chamber at PSI was $4.0 \pm 0.3$ slpm. Decays of the $HO_2$ radical along an aerosol
flow tube were measured using a Fluorescence Assay by Gas Expansion (FAGE) detector in both the
absence and presence of different concentrations of aerosol particles. All experiments were performed
at room temperature ($293 \pm 2$ K).
The HO$_2$ radical was formed via Reactions 1 – 2, by passing a humidified flow over a mercury penray
lamp (L.O.T. Oriel, model 6035) in the presence of trace amounts (20 – 30 ppm) of oxygen in the
nitrogen flow.

H$_2$O + $hv$ → OH + H                                                                                    (R1)
H + O$_2$ + M → HO$_2$ + M                                                                                (R2)

Data acquisition was only started once HO$_2$ concentrations within the flow tube were stable
which occurred within 1 minute of switching on the mercury lamp. The HO$_2$ radicals entered the FAGE
cell through a 0.7 mm diameter pinhole, and were then converted to OH by reacting with added NO.
The FAGE cell was either kept at a pressure of ~ 0.85 Torr or ~ 1.5 Torr using a combination of a rotary
pump (Edwards, model E1M80) and a roots blower (EH1200). The OH radicals were detected by laser
induced fluorescence at 308 nm (Heard and Pilling, 2003;Stone et al., 2012). Initial HO$_2$ concentrations
(obtained by calibration) exiting the injector were measured as ~ 1 × 10$^9$ molecule cm$^{-3}$ for all
experiments (following mixing and dilution with the main flow), and the concentration was then
measured as a function of distance along the flow tube.
For the experiments using copper doped sucrose aerosol particles, 3.42 grams of sucrose
(Fisher, > 99%) and 0.125 grams of copper (II) sulphate pentahydrate were dissolved in 500 ml of
milliQ water. These solutions were then placed in an atomiser (TSI, 3076) in order to form aerosol
particles. The aerosol particles passed through a neutraliser (Grimm 5522) and an impactor before
entering the conditioning flow tube. The size distribution of the aerosol particles were then measured
at the end of the reaction flow tube using a Scanning Mobility Particle Sizer (SMPS, TSI, 3080).
The experimental setup used to measure previous HO$_2$ uptake coefficients (George et al.,
2013;Matthews et al., 2014;Lakey et al., 2015a;Lakey et al., 2015b) was transported from the University
of Leeds, UK, to the Paul Scherrer Institute, Switzerland, where it was connected to the Paul Scherrer
Institute (PSI) smog chamber and, for some of the experiments, also to a Potential Aerosol Mass (PAM)
chamber (see Figure 1). The PSI smog chamber has a volume of 27 cubic metres, it is made from 125
μm Teflon fluorocarbon film and has been described elsewhere (Paulsen et al., 2005). To initiate
photochemical reactions four 4 kW xenon arc lamps (light spectrum >280 nm, OSRAM) and eighty
black lights (100W tubes, light spectrum between 320 and 400 nm, Cleo Performance) were used. For
most experiments the chamber was first humidified to 50% relative humidity, but for two experiments
this was increased to 80%, after which the precursor gases were added. The concept, design and
operation of a PAM chamber has also previously been described (Kang et al., 2007). The PAM chamber
at PSI is a flow tube of 0.46 m in length and 0.22 m internal diameter. Two low pressure Hg lamps
mainly emitting at 185 and 254 nm produce ozone in the chamber. Water vapour was photolysed by the
185 nm radiation to produce OH and $HO_2$ and also photolysed $O_2$ to produce $O_3$, whereas the 254 nm
light could also photolyse $O_3$ to produce OH following the reaction of $O(^1D)$ with water vapour. Upper-
limit OH production rates are in the range of $1 \times 10^{12}$ - $2 \times 10^{12}$ molecule $cm^{-3}$ $s^{-1}$ (Bruns et al., 2015).
The composition and oxidation state of SOA formed within PAM chambers has previously been shown
to be similar to SOA generated within environmental chambers (Bruns et al., 2015;Lambe et al., 2011a)
and SOA in the atmosphere (Ortega et al., 2015).

Four different types of experiments were performed.

(i) α-pinene ozonolysis in the PSI smog chamber (600 ppb α-pinene, 280 ppb ozone: ozone was added
first to the chamber; after injection of α -pinene particle nucleation and growth rapidly occurred).
(ii) OH initiated α-pinene photochemistry in the smog chamber (500 ppb α-pinene, 350 ppb $NO_2$: Xenon
and black lights where used to initiate photochemical reactions).
(iii) OH initiated α-pinene photochemistry in the PAM chamber (500 ppb α-pinene was filled into the
large smog chamber at 50 or 80 % RH to supply a constant concentration of α-pinene to the PAM
chamber, all SOA was formed within the PAM chamber).
(iv) OH initiated TMB photochemistry in the PAM chamber (2 ppm TMB was filled into the large smog
chamber at 50 % RH to supply a constant concentration of TMB to the PAM chamber, all SOA was
formed within the PAM chamber).

These precursor concentrations were chosen in order to obtain a large enough aerosol surface

area in the flow tube to be able to measure a $HO_2$ uptake coefficient. Experiments were performed only
once the aerosol surface area within the aerosol flow tube exceeded $5 \times 10^{-5}$ $cm^2$ $cm^{-3}$, and in the case
of the smog chamber experiments once a maximum aerosol concentration had been reached (as
summarised in the Results Section). Prior to entering the flow tube, the aerosol flow from the smog or
PAM chamber (4.0 slpm) was passed through either two or three cobalt oxide denuders in series (each
40 cm long, 0.8 cm inner diameter quartz tubes coated with cobalt oxide prepared by thermal
decomposition of a saturated $Co(NO_3)_2$ solution applied to its inner walls at 700°C as described in
Ammann (2001)), which in turn were in series with a charcoal denuder (length = 16.4 cm, diameter =
0.9 cm, 69 quadratic channels) in order to remove $NO_x$ species, $RO_2$, VOC's and ozone that had been
present in the chamber. These denuders have previously been shown to be extremely efficient at
removing gas phase $NO_x$ and VOCs (Arens et al., 2001). It should be noted that the flows were drawn
through the aerosol flow tube using a pump instead of the normal procedure whereby the flows are
pushed through the experimental setup using mass flow controllers. The pumping setup led to slightly
reduced pressures (904 – 987 mbar) in the aerosol flow tube, and so careful checks were performed to
ensure that the flow tube was vacuum tight. The aerosol size distribution from which the surface area
exiting the flow tube was calculated was measured using a Scanning Mobility Particle Sizer (SMPS),
which consisted of a neutraliser (Kr-85), a Differential Mobility Analyser (DMA, length 93.5 cm, inner
radius 0.937 cm and outer radius 1.961 cm) and a CPC (TSI, model 3022). A typical surface weighted
aerosol size distribution for the α-pinene derived aerosol particles is shown in Figure 2. Note that an
impactor was not used in the experimental setup for the SOA measurements as this restricted the flow
that could be pumped through the flow tube and was also found to be unnecessary as the aerosol size
distribution from the chambers fell entirely within the range of aerosol sizes that the SMPS could
measure.
In order to check that the experimental setup used at PSI produced consistent results with those
previously performed at the University of Leeds, an experiment was performed with ammonium
sulphate aerosol particles. The ammonium sulphate aerosol particles were formed using an atomiser
rather than aerosol particles being formed in a chamber, but were then passed through the same set up
(including the denuders) as the SOA was passed through. The experiment was performed at a flow tube
pressure of 915 mbar, due to the flows being pumped through the setup, (compared to pressures of 904
– 987 mbar for the SOA experiments), and a $HO_2$ uptake coefficient of 0.004 ± 0.002 was measured at
60% RH which is in agreement with previous experiments by George et al. (2013), which were
performed at atmospheric pressure (~ 970 – 1040 mbar).

## Data analysis


Experiments were performed by moving the $HO_2$ injector backwards and forwards along the
flow tube either in the presence of or in the absence of aerosol particles, and recording the FAGE signal
from $HO_2$ radicals. The background signal in the absence of $HO_2$ (mercury lamp in the injector switched
off), but with the NO entering the FAGE cell, was recorded and was subtracted, from the signal during
experiments. For α-pinene experiments this background signal was small and similar to previous
experiments using dust, organic and inorganic salt aerosol particles (George et al., 2013;Lakey et al.,
2015b;Lakey et al., 2015a;Matthews et al., 2014). However, for the TMB experiments this background
signal varied from about half to two thirds of the signal from $HO_2$ with the mercury lamp in the injector
switched on. The background signal disappeared when the NO added to the FAGE cell was switched
off showing that it was not due to OH. The background signal within experiments did not change when
aerosol particles were present compared to when they were completely filtered out (see Figure 1).
Although the denuders are efficient at removing gas phase species (Arens et al., 2001), it can be

hypothesized that the signal was due to the formation of $HO_2$ and $RO_2$ radicals generated by a small fraction of ozone, precursors and oxidation products passing through the denuders for the TMB experiments. $RO_2$ species would have been observed as a $HO_2$ interference by the FAGE detection method. FAGE interferences have previously been observed for alkene, aromatic and > C3 alkane derived $RO_2$ (Fuchs et al., 2011;Whalley et al., 2013). A box model was run, utilising chemistry within the Master Chemical Mechanism (MCM 3.2), which is detailed further in Whalley et al. (2013)), and constrained to the experimental concentrations, and showed that the expected interference from TMB $RO_2$ and α-pinene $RO_2$ would have been equivalent to $0.59 \times [HO_2]$ and $0.44 \times [HO_2]$, respectively, at a NO flow of 50 ml min$^{-1}$ into the FAGE cell, a FAGE pressure of 1.5 Torr and a flow through the FAGE pinhole of 4.2 slpm. However, for α-pinene experiments the background signal did not change between the NO being switched on and off with the mercury lamp switched off in the injector, indicating the absence of interferences in the FAGE cell for these experiments. The lack of interference for the α-pinene experiments suggests that the denuders were more efficient at removing the gas phase precursors and oxidation products from the chamber and that only negligible concentrations of $RO_2$ species were present in the flow tube. Nevertheless, since for the TMB experiments a significant background signal was observed, that signal was measured regularly throughout the experiment and used to correct the measurement data.

HO$_2$ decays along the flow tube in the presence and absence of aerosol particles were measured between ~ 10 and 18 seconds flow time after the point of injection to ensure thorough mixing. A previous calculation showed that the flows should be fully mixed by ~ 7 seconds (George et al., 2013). An example of the HO$_2$ decays in the presence and absence of aerosol particles for a TMB experiment is shown in Figure 3, plotted as the natural logarithm of HO$_2$ signal (proportional to concentration) against reaction time according to:

$$ln\frac{[HO_2]_t}{[HO_2]_0} = -k_{obs}t \tag{E1}$$

There is clear uptake of HO$_2$ observed by the SOA derived from TMB. The pseudo first-order rate coefficients ($k_{obs}$) were then corrected for wall losses and non-plug flow conditions using the methodology described by Brown (1978). The average correction was 22%. These corrected rate constants ($k'$) were related to the HO$_2$ uptake coefficient ($\gamma_{obs}$) by the following equation:

$$k' = \frac{\gamma_{obs}\omega_{HO2}S}{4} \tag{E2}$$

where $\omega_{HO2}$ is the molecular thermal speed of HO$_2$ and $S$ is the total aerosol surface area. Examples of $k'$ as a function of the aerosol surface area is shown in Figure 4. The HO$_2$ uptake coefficients were then corrected for gas-phase diffusion limitations using the methodology described by (Fuchs and Sutugin, 1970), although this correction changed the uptake coefficient by less than 1 % for all experiments.


## Model description


The kinetic multi-layer model of aerosol surface and bulk chemistry (KM-SUB) has been
described in detail by Shiraiwa et al. (2010). It is a multi-layer model comprising a gas phase, a near-
surface gas phase, a sorption layer, a near-surface bulk layer and a number of bulk layers arranged in
spherical geometry. Processes that can occur within the model include gas-phase diffusion, adsorption
and desorption, bulk diffusion, and chemical reactions in the gas phase, at the surface and in the bulk.
In contrast to traditional resistor models, the KM-SUB model enables efficient treatment of complex
chemical mechanisms. Input parameters to the model are summarised in Table 1 whilst the reactions
that were included are shown below:

$$HO_{2(g)} + HO_{2(g)} \rightarrow H_2O_{2(g)} + O_{2(g)} \qquad\qquad k_{GP} \qquad\qquad (R3)$$


$$HO_{2(aq)} \rightleftharpoons H^+_{(aq)} + O_2^-_{(aq)} \qquad\qquad K_{eq} \qquad\qquad (R4)$$

$$HO_{2(aq)} + HO_{2(aq)} \rightarrow H_2O_{2(aq)} + O_{2(aq)} \qquad\qquad k_{BR,1} \qquad\qquad (R5)$$

$$HO_{2(aq)} + O_2^-_{(aq)} + H_2O_{(l)} \rightarrow H_2O_{2(aq)} + O_{2(aq)} + OH^-_{(aq)} \qquad\qquad k_{BR,2} \qquad\qquad (R6)$$


$$Cu^{2+}_{(aq)} + HO_{2(aq)} \rightarrow O_{2(aq)} + Cu^+_{(aq)} + H^+_{(aq)} \qquad\qquad k_{BR,3} \qquad\qquad (R7)$$

$$Cu^{2+}_{(aq)} + O_2^-_{(aq)} \rightarrow O_{2(aq)} + Cu^+_{(aq)} \qquad\qquad k_{BR,4} \qquad\qquad (R8)$$

$$Cu^+_{(aq)} + HO_{2(aq)} + H_2O_{(l)} \rightarrow H_2O_{2(aq)} + Cu^{2+}_{(aq)} + OH^-_{(aq)} \qquad\qquad k_{BR,5} \qquad\qquad (R9)$$

$$Cu^+_{(aq)} + O_2^-_{(aq)} + 2H_2O_{(l)} \rightarrow H_2O_{2(aq)} + Cu^{2+}_{(aq)} + 2OH^-_{(aq)} \qquad\qquad k_{BR,6} \qquad\qquad (R10)$$


The bulk layer number was set to 100 corresponding to a bulk layer thickness of 0.5 nm which
is only slightly larger than the diameter of $HO_2$ (0.4 nm) and implies that $HO_2$ only needs to travel
approximately the distance of its own diameter to go from being an adsorbed radical on the surface of
the aerosol particle to a dissolved aqueous radical. The same short distance must be overcome by $HO_2$
to move between bulk layers, which is important for convergence of the numerical model, especially
when the chemical reactions within the aerosol particles are very fast compared to the diffusion time
scales, leading to steep concentration gradients within the particle. Reducing the bulk layer thickness
further did not significantly impact the calculated uptake coefficients.
During experiments the average radius was observed to change by less than 10 % over the range
of humidities, and therefore an assumption was made within the model that the average aerosol radius
remained constant over the range of relative humidities. For the diffusion coefficient of $HO_2$ within
aerosol particles we used the measured diffusion coefficients of $H_2O$ within sucrose solutions, which
we then corrected using the Stokes-Einstein equation to take into account the larger radius of $HO_2$
radicals compared to $H_2O$ molecules (Price et al., 2014;Zobrist et al., 2011). The correction resulted in
a factor of 1.22 decrease in the diffusion coefficients of $HO_2$ compared to the diffusion coefficients of
$H_2O$. It should be noted that above a viscosity of 10 Pa s the Stokes-Einstein relationship starts to fail
and that the effect of increasing molecular size may become much stronger (Power et al., 2013). Price
et al. (2014) estimated diffusion coefficients of $H_2O$ by using Raman spectroscopy to observe $D_2O$
diffusion in high-viscosity sucrose solutions whilst Zobrist et al. (2011) used optical techniques to
observe changes in the size of sucrose particles when exposed to different relative humidities.
Sensitivity tests showed that the diffusion rate constants of $O_2^-$, $Cu^+$ and $Cu^{2+}$ did not influence
calculation results. The reaction rate coefficients involving copper ($k_{BR,3}$ - $k_{BR,6}$) are so large that $O_2^-$ is
produced *in situ* and consumed locally. The catalytic nature of these reactions cause $Cu^+$ and $Cu^{2+}$ to
rapidly interconvert meaning that they remain available at high concentrations in the upper layers of the
aerosol particle. Similarly, as sucrose does not react with any species within the model, its diffusion
within the model is unimportant to the outputted $HO_2$ uptake coefficient.

## Results and Discussion


### $HO_2$ uptake by copper doped sucrose aerosol particles


The results of the $HO_2$ uptake coefficient measurements onto copper doped sucrose aerosol
particles as a function of relative humidity (RH) are shown in Figure 5. The results show a large
dependence upon relative humidity with the $HO_2$ uptake coefficient increasing from $0.012 \pm 0.007$ at
$17 \pm 2$ % RH to $0.22 \pm 0.06$ at relative humidities above 65%. The latter value is likely equal to the
surface accommodation coefficient, and is consistent with many previous studies (Takahama and
Russell, 2011;George et al., 2013;Lakey et al., 2015b). At lower humidities, the diffusion coefficients
decrease which leads to slower transport of $HO_2$ within the bulk, and therefore to a slower overall rate
of $HO_2$ destruction (Reactions 7 – 10). The $HO_2$ reacto-diffusive length (Hanson et al., 1994;Schwartz
and Freiberg, 1981) varied from between ~ 4 – 7 nm at the highest relative humidity that was used (71
% RH) down to ~0.006 – 0.05 nm at the lowest relative humidity (17 % RH). The range of values for
the reacto-diffusive length at a given RH is due to the difference between the parameterizations of the
diffusion coefficient in Price et al. (2014) and Zobrist et al. (2011). These reacto-diffusive lengths
indicate that at all relative humidities $HO_2$ radicals will be limited to the outermost molecular layers of
the particle before reacting away, which is in agreement with the model. Note that it was shown in
previously that the uptake of gas-phase species generally increases with increasing reacto-diffusive
length, which is consistent with our $HO_2$ uptake coefficient measurements (Slade and Knopf,
2014;Davies and Wilson, 2015;Houle et al., 2015). The red and blue lines in Figure 5 show the predicted
$HO_2$ uptake coefficients using the KM-SUB model when using two different parameterisations for $HO_2$
diffusion coefficients as a function of RH (see the model description). There is good agreement between
the model and the measurements suggesting that the change in $HO_2$ uptake over the range of humidities
is indeed due to a change in the $HO_2$ diffusion coefficient which is in turn due to a change in the viscosity
of the aerosol particles. Sensitivity tests showed that an increase in the rate constants of reactions R7 –
R10 does not affect the $HO_2$ uptake coefficient. A two order of magnitude decrease in the rate constants
affects the uptake coefficient marginally by reducing it by less than 10 % in the 40 – 55 % relative
humidity range, but has no impact at the lower or higher relative humidities.

Using the kinetic framework and classification scheme of Berkemeier et al. (2013), Figure 6

illustrates how the change in relative humidity leads to a change in the kinetic regime of $HO_2$ uptake.
At the highest relative humidities the uptake is limited by surface accommodation. At intermediate
relative humidities with $\gamma < \alpha_{s,0}$, the uptake is limited by surface-to-bulk transport, which is related to
both solubility (Henry's law coefficient) and diffusivity (diffusion coefficient) in the kinetic model.
Under both conditions, the uptake is driven by chemical reaction in the near-surface bulk and effectively
limited by mass accommodation, which includes both surface accommodation and surface-to-bulk
transport (Behr et al., 2009;Berkemeier et al., 2013). At low relative humidities the $HO_2$ uptake
coefficient was limited by chemical reaction at the surface as discussed below (Berkemeier et al., 2013).

Although the viscosity changes by more than 8 orders of magnitude and the diffusion

coefficients change by 5-7 orders of magnitude over the investigated range of relative humidity, the
measured $HO_2$ uptake coefficients change by only ~ 1 order of magnitude. This can be explained to
some extent by the uptake coefficient being proportional to the square root of the diffusion coefficient
when the uptake is controlled by reaction and diffusion of $HO_2$ in the bulk (Davidovits et al.,
2006;Berkemeier et al., 2013). If this were the only mechanism involved, however, one would still
expect a change in the uptake coefficient by 2.5 – 3.5 orders of magnitude. The most plausible
explanation for the relatively high $HO_2$ uptake coefficients observed at low relative humidities is a
surface reaction of $HO_2$. For example, at 17 % RH and without a surface reaction, $\gamma$ values as low as ~5
$\times\ 10^{-4}$ and ~3 $\times\ 10^{-5}$ would be expected using the Zobrist et al. (2011) and Price et al. (2014)
parameterisations, respectively. However, by including the following self-reaction of $HO_2$ at the surface
of the sucrose particles, much better agreement with the observed values of around ~$10^{-2}$ could be
obtained (Fig. 5):

$$HO_2 + HO_2 \xrightarrow{Cu^{2+/+}} H_2O_2 + O_2 \qquad k_{Surf} = 1 \times 10^{-8} \text{ cm}^2 \text{ s}^{-1} \qquad \text{(R11)}$$

Although the true mechanism for reaction at the surface remains unclear, the large rate constant for this
reaction suggests that copper could potentially be catalyzing the destruction of $HO_2$ at the surface of the
sucrose particles which is consistent with the higher $HO_2$ uptake coefficients measured onto solid
aerosol particles containing transition metals compared to solid aerosol particles containing no
transition metal ions (Matthews et al., 2014;Lakey et al., 2015a;Bedjanian et al., 2013;George et al.,
2013). Note however that for a relevant surface reaction in kinetic flux models, it is necessary to use an
effective desorption lifetime $\tau_d$ in the millisecond to second time range (Berkemeier et al.,
2016;Shiraiwa et al., 2010). This is many orders of magnitude longer than would be expected due to
pure physisorption as estimated by molecular dynamic simulations (Vieceli et al., 2005), indicating that
the adsorption process should involve chemisorption or formation of long-lived intermediates that
would have the potential to extend these effective desorption lifetimes (Shiraiwa et al.,
2011a;Berkemeier et al., 2016). The effect and importance of surface reactions is consistent with
previous work by Gržinić et al. (2015), Steimer et al. (2015) and Berkemeier et al. (2016) for the uptake
of $N_2O_5$ to citric acid and the uptake of $O_3$ to shikimic acid over a range of relative humidities. A second
potential reason for the discrepancy at low humidities could be an incomplete equilibration of the
aerosol particles with respect to RH, as they had only been mixed with the conditioning flow for ~ 10
seconds before entering the reaction flow tube. Bones et al. (2012) inferred from measurements on
larger particles that for 100 nm diameter sucrose aerosol particles the equilibration time would be more
than 10 seconds when the viscosity increased above ~ $10^5$ Pa s, which would occur at ~ 43 % RH (Power
et al., 2013). The actual diffusion coefficients would thus be higher than assumed in calculations which
assume fully equilibrated particles. However, the near-surface bulk of the aerosol particles, where the
reactions occur, would be much better equilibrated with respect to RH than the inner core of the aerosol
particles (Berkemeier et al., 2014). This means that the lack of aerosol equilibration with respect to RH
is likely to have a negligible impact upon the $HO_2$ uptake coefficient.
It should also be noted that the KM-SUB modelling results were very sensitive to the initial
aerosol pH. For example, at a pH of 4.1 (used in Figure 5, the reason for this value is discussed below)
the $HO_2$ uptake coefficient as predicted by the KM-SUB model at 50 % RH (using the Zobrist et al.
(2011) $H_2O$ diffusion coefficients) was $\gamma = 0.06$ compared to $\gamma = 0.11$ at pH 5 and $\gamma = 0.21$ at pH 7. The
reason for this strong dependence upon pH has been discussed previously and is due to the partitioning
of $HO_2$ with its conjugate base $O_2^-$, as shown by Reaction 4, affecting the effective Henry's law
coefficient and the effective rate constants (Thornton et al., 2008). Although it was not possible to
measure the actual pH of the aerosol particles, it was possible to estimate the concentration of copper
(II) sulphate (which is a weak acid) within the aerosol particles using the known growth factors of
sucrose aerosol particles (Lu et al., 2014). The pH of 0.05 M and 0.1 M copper (II) sulphate solutions
(which were calculated to be the extremes of the possible copper concentrations over the RH range)
were then measured using a pH meter (Jenway, 3310) as being in the range of $4.10 \pm 0.05$. It is expected
that the pH would be dominated by the presence of copper sulfate rather than sucrose which has a pH
of 7 in water and a very high pKa of 12.6. Therefore, there is confidence that the correct initial aerosol
pH was inputted into the model. Hence, while the $HO_2$ uptake coefficient might depend on further
factors such as aerosol pH, a clear dependence on relative humidity, and hence particle viscosity could
be observed, and it remains likely that at low humidity a surface loss process becomes dominating.

**$HO_2$ uptake by SOA**

A summary of all $HO_2$ uptake experiments performed on SOA is shown in Table 2. On average
the $HO_2$ uptake coefficient was measured as $0.004 \pm 0.002$ onto TMB derived aerosol particles produced
in the PAM chamber, whereas for α-pinene derived aerosol particles only an upper limit of 0.001
(obtained from the error in the slope of Figure 4(a)) could be placed on the $HO_2$ uptake coefficient at
50 and 80 % RH. It should be noted that for the α-pinene experiments the $HO_2$ uptake coefficient was
non-measurable for both ozonolysis and photochemistry experiments using both the smog chamber and
the PAM chamber as sources of the SOA, and therefore only upper limits of individual experiments are
reported in

Table 2. There was some variability for the upper limits that were measured for individual α-
pinene experiments which is likely to be due to the maximum aerosol surface-to-volume ratio that was
obtained in each experiment.
There are several possible reasons for the larger $HO_2$ uptake coefficients being measured for
the TMB derived aerosol particles compared to the α-pinene derived aerosol particles. These reasons
will be summarised below, but include a differing particle viscosity, a different particle liquid water
content or a $HO_2 + RO_2$ reaction occurring within the aerosol particles. Although the viscosity of α-
pinene derived aerosol has been measured as $\sim 10^3$ Pa s at 70 % RH and $> 10^9$ Pa s for RH < 30 %, to
our knowledge, there are currently no measurements of the viscosity of TMB derived aerosol published
in the literature (Renbaum-Wolff et al., 2013). By running the KM-SUB model it can be estimated that
the diffusion coefficient of $HO_2$ within the particles would need to be approximately $1 \times 10^{-10}$ cm$^2$ s$^{-1}$ for
TMB derived aerosol particles and $< 5 \times 10^{-12}$ cm$^2$ s$^{-1}$ for α-pinene derived aerosol particles. This range
of values seems to be consistent with the diffusion coefficients estimated by Berkemeier et al. (2014)
and Lienhard et al. (2015) for water diffusion in low and medium O:C SOA.

Thornton et al. (2003) previously suggested that for malonic acid aerosol particles the liquid
water content could be limiting the aqueous chemistry below 40 % RH. As can be seen by the $HO_2$
reaction scheme, the rate of Reaction R6 is dependent upon the liquid water concentration within the
aerosol, and therefore the uptake coefficient could be limited by a low aerosol liquid water content.
However, there remains some uncertainty as to whether the liquid water content of TMB derived aerosol
particles would be higher than the liquid water content of α-pinene derived aerosol particles. Duplissy
et al. (2011) measured a higher hygroscopicity parameter ($\kappa_{org}$) for TMB derived aerosol particles
compared to α-pinene derived aerosol particles whereas Lambe et al. (2011b) and Berkemeier et al.
(2014) stated the opposite. However, as well as being dependent upon the hygroscopicity parameter,
the liquid water content of the aerosol particles would also be dependent upon the O:C ratio in the SOA.

If the viscosity and liquid water content of the α-pinene and TMB derived aerosol particles are
similar, the larger $HO_2$ uptake coefficients measured for TMB derived aerosol particles could be due to
a higher reactivity of these aerosol particles towards $HO_2$. This could be the case if the TMB derived
aerosol particles contained reactive radical species such as organic peroxy radicals, $RO_2$, which partition
into the aerosol or are formed within the aerosols by intra-particular reactions (Donahue et al., 2012;Lee
et al., 2016). As previously stated in the Data Analysis section, during α-pinene experiments, no
indication of $RO_2$ being present in the flow tube was observed by FAGE as a $HO_2$ interference.
However, for TMB derived aerosol particles, a large background signal was observed by FAGE
indicating that reactive radical species were likely to be present within the flow tube. If the reaction of
$HO_2$ with these species at the surface or within the bulk of the aerosol was faster than the equivalent
gas phase reaction, a larger $HO_2$ uptake coefficient would be observed.

## Atmospheric implications and conclusions


The effect of aerosol viscosity upon $HO_2$ uptake coefficients was systematically investigated
with a combination of $HO_2$ uptake coefficient measurements and a state-of-the-art kinetic model. A
good correlation was obtained between measured $HO_2$ uptake coefficients onto copper doped sucrose
aerosols as a function of RH and the KM-SUB model output. At higher relative humidities the uptake
was limited by mass accommodation whilst at lower relative humidities the aerosol particles were
viscous and the uptake was limited by surface reaction. These results imply that viscous aerosol particles
will have very little impact upon gaseous tropospheric $HO_2$ concentrations.

The first measurements of the $HO_2$ uptake coefficient onto SOA have been reported in this
work. The $HO_2$ uptake coefficient measured for α-pinene derived aerosol particles was below the limit
of detection of the apparatus ($\gamma < 0.001$) whereas for TMB derived aerosol particles the uptake
coefficient was measurable ($\gamma = 0.004 \pm 0.002$). These results are consistent with the copper doped
sucrose results, and indicate that the impact of SOA on gaseous $HO_2$ concentrations would likely be
small. However, it remains unclear as to the reasons for the larger $HO_2$ uptake coefficient measured
onto TMB derived aerosol particles compared to α-pinene derived aerosol particles. The possibility that
the larger uptake coefficient onto TMB derived aerosol particles was due to a lower viscosity of the
aerosol particles or a higher liquid water content compared to α-pinene derived aerosol particles cannot
be confirmed until further measurements of the viscosity and liquid water content of TMB derived
aerosol particles are published in the literature. However, if the larger uptake coefficients are due to a
$HO_2 + RO_2$ reaction within the aerosol, this could impact the $HO_2$ uptake coefficient for any aerosol
containing $RO_2$. The actual increase would depend on a variety of factors such as the concentrations of
$RO_2$, the partition coefficients of $RO_2$ to the aerosol particles, the reactivity of different $RO_2$ species
with $HO_2$ radicals and the intra-particular formation of $RO_2$ and other reactive radicals (Lee et al.,
2016;Donahue et al., 2012;Tong et al., 2016). The $HO_2 + RO_2$ reaction could potentially occur within
the majority of aerosol particles within the atmosphere, this could have implications for the gaseous
$HO_2$ and $RO_2$ concentrations in the troposphere which could then impact upon the concentrations of
other species such as ozone.

## Acknowledgements


PSJL is grateful to NERC for the award of a studentship. LKW and DEH are also grateful to the NERC
funded National Centre for Atmospheric Science for ongoing support and to NERC for funding of the
$HO_2$ aerosol uptake apparatus (grant reference NE/F020651/1). TB was supported by the Max Planck
Graduate Center with the Johannes Gutenberg-Universität Mainz (MPGC). The experiments at PSI
were supported by T. Bartels-Rausch and M. Birrer. MA and MK were supported by the Swiss National
Science Foundation (grant nos 149492, CR3213-140851).





**Figures**

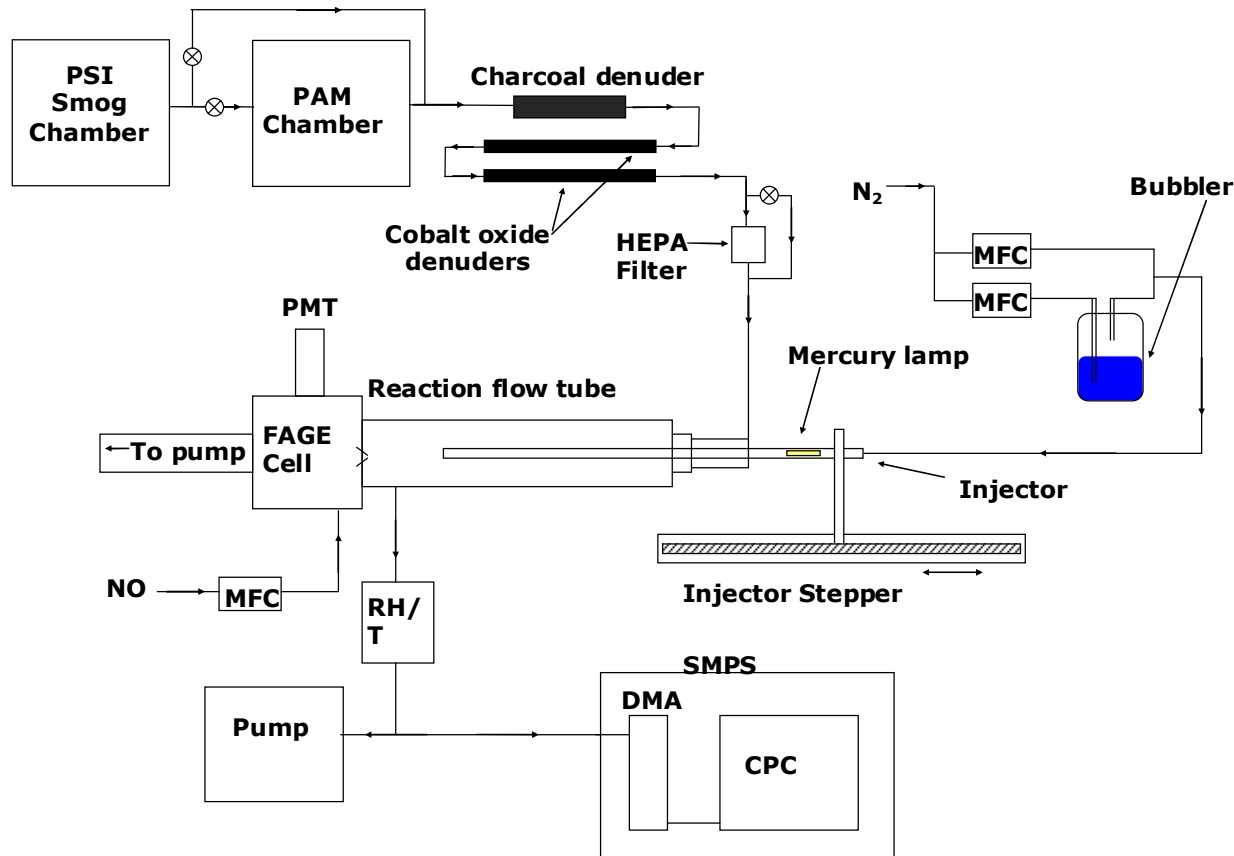


**Figure 1:** A schematic of the experimental setup used to measure $HO_2$ uptake coefficients onto SOA
aerosol particles. Key: PAM- Potential aerosol mass, PMT- Photomultiplier tube, FAGE-
Fluorescence Assay by Gas Expansion, MFC- Mass flow controller, RH/ T- relative humidity and
temperature probe, SMPS- Scanning mobility particle sizer, DMA- Differential mobility analyser,
CPC- Condensation particle counter.



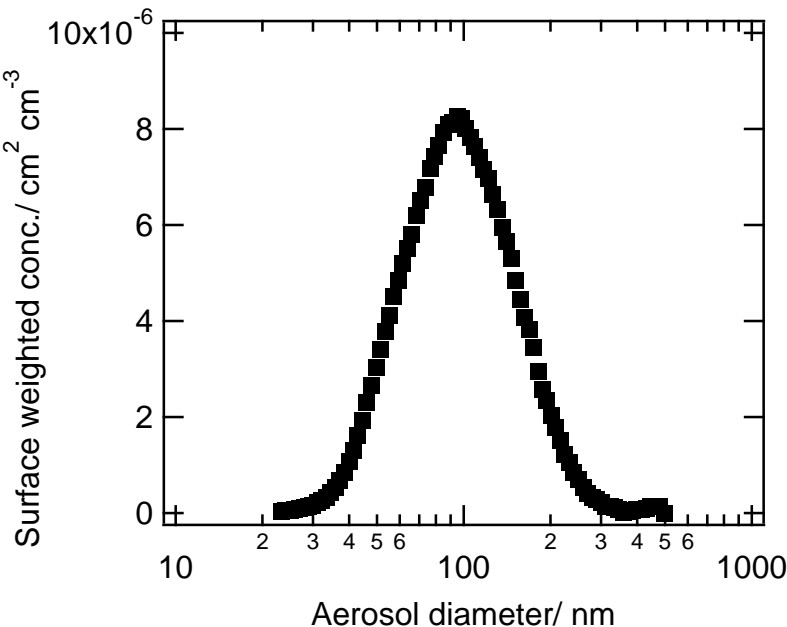


**Figure 2:** An example of the size distribution for α-pinene derived aerosol particles formed in the
PAM chamber at a relative humidity of ~ 50 %.
















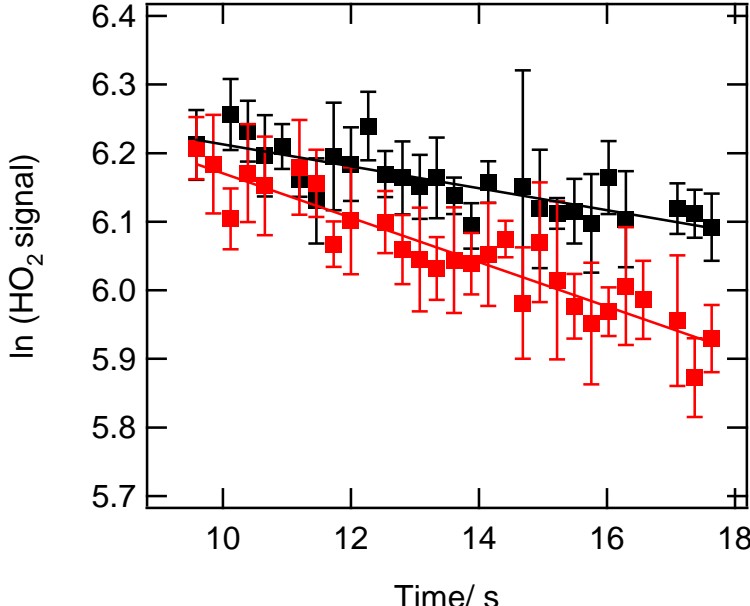


**Figure 3:** Examples of the HO₂ wall loss without any aerosol particles along the flow tube (black squares) and the HO₂ loss with an aerosol surface area of $2.2 \times 10^{-4}$ cm² cm⁻³ for TMB derived aerosol particles at an initial HO₂ concentration of ~ $1 \times 10^{9}$ molecule cm⁻³ (red squares) and for RH = 50 %. The error bars represent one standard deviation in the measured HO₂ signal for a measurement time per point of 3 seconds.











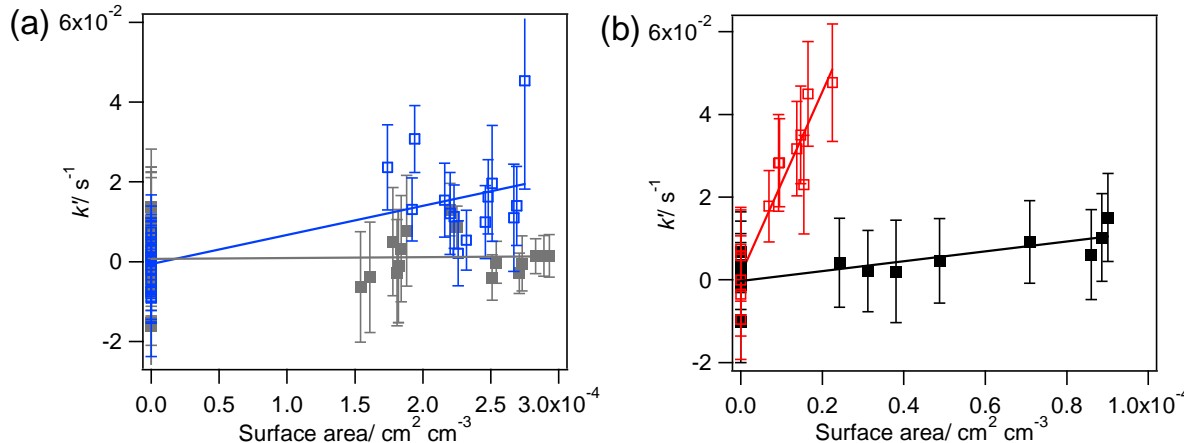



**Figure 4:** The pseudo-first-order rate constants with the wall losses subtracted as a function of aerosol surface area for (a) α-pinene derived aerosol particles (grey) and TMB derived aerosol particles (blue) at 50 % RH and a pressure of 904 – 929 mbar and (b) copper doped sucrose aerosol particles at 17% RH (black) and 71% RH (red) at atmospheric pressure. Experiments were performed at 293 ± 2 K. In panel (a) experiments were performed using the PAM chamber as the source of aerosol particles and represent experiments 5 and 6 in Table 2. Error bars represent the 1 standard deviation propagated uncertainty for individual determinations of $k'$. The data points at an aerosol surface area of 0 $cm^2$ $cm^{-3}$ (no aerosol particles present) are repeats of the wall loss decays taken throughout the experiment and are within error of each other.






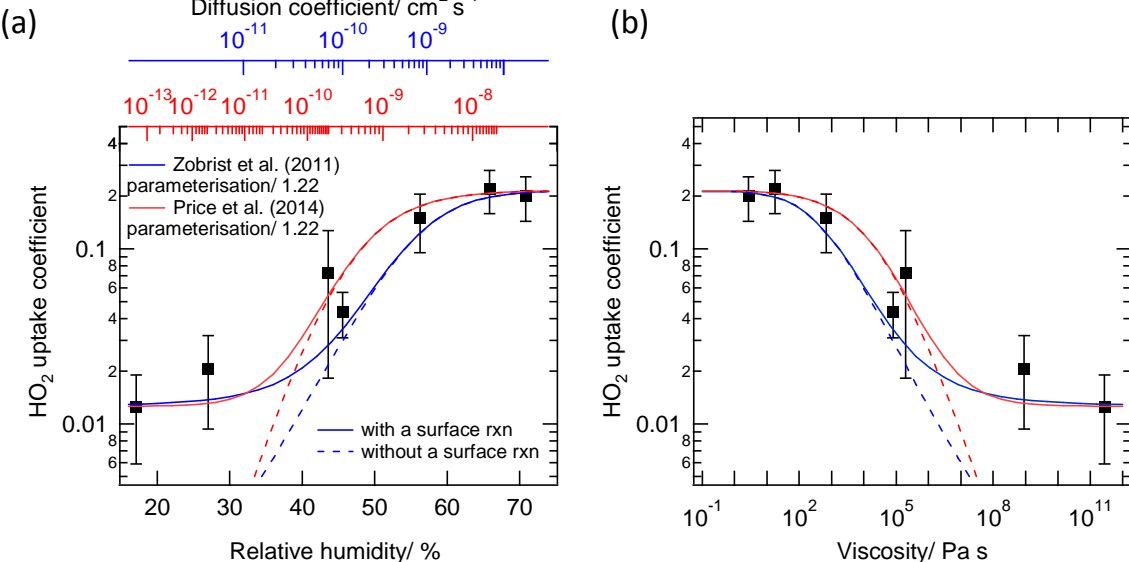


**Figure 5:** The HO$_2$ uptake coefficient onto copper (II) doped sucrose aerosol particles as a function of (a) relative humidity and (b) aerosol particle viscosity. The lines represent the expected HO$_2$ uptake coefficient calculated using the KM-SUB model using the Price et al. (2014) (red) and Zobrist et al. (2011) (blue) diffusion parameterisations (see model description section) and with (solid) and without (dashed) the inclusion of a surface reaction (Reaction R11). The viscosity within sucrose aerosol particles is based upon the data and fitting shown in Power et al. (2013) and Marshall et al. (2016) whilst the red and blue axes in panel (a) are the Price et al. (2014) and Zobrist et al. (2011) diffusion parameterisations, respectively. The error bars represent two standard deviations of the propagated error in the gradient of the $k'$ against aerosol surface area graphs.





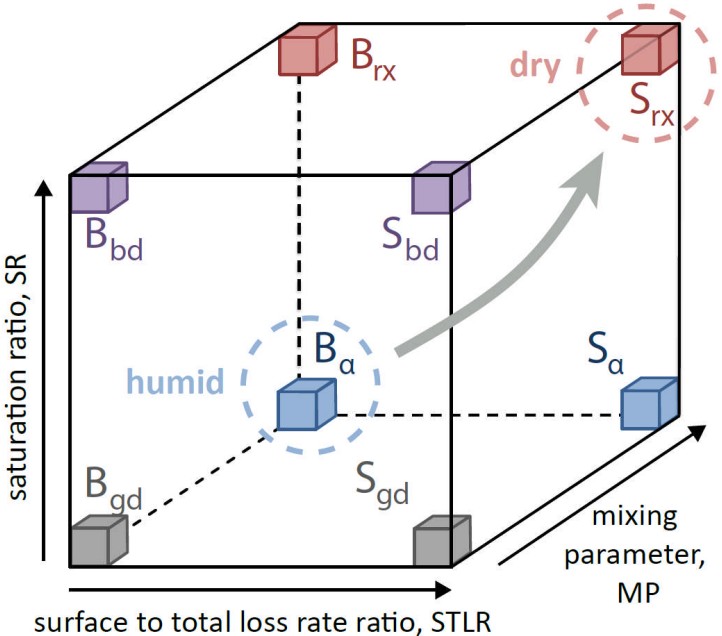


**Figure 6:** The kinetic cube representing the eight limiting cases for uptake of gases to aerosol particles (Berkemeier et al., 2013). $B_{rx}$: bulk reaction limited by chemical reaction, $B_{bd}$: bulk reaction limited by bulk diffusion of the volatile reactant and the condensed reactant, $B_\alpha$: bulk reaction limited by mass accommodation, $B_{gd}$: bulk reaction limited by gas-phase diffusion; $S_{rx}$: surface reaction limited by chemical reaction, $S_{bd}$: surface reaction limited by bulk diffusion of a condensed reactant, $S_\alpha$: surface reaction limited by surface accommodation, $S_{gd}$: surface reaction limited by gas-phase diffusion. For copper doped sucrose aerosol particles, the $HO_2$ uptake coefficient is limited by mass accommodation under humid conditions and by chemical reaction at the surface at low relative humidity.













# Tables



**Table 1:** The parameters used in the KM-SUB $HO_2$ uptake model over all relative humidities.

| Parameter | Description | Value at 293 K | Reference |
|---|---|---|---|
| $k_{BR,1}$ | Rate constant, R5 | $1.3 \times 10^{-15}$ cm$^3$ s$^{-1}$ | Thornton et al. (2008) |
| $k_{BR,2}$ | Rate constant, R6 | $1.5 \times 10^{-13}$ cm$^3$ s$^{-1}$ | Thornton et al. (2008) |
| $k_{BR,3}$ | Rate constant, R7 | $1.7 \times 10^{-13}$ cm$^3$ s$^{-1}$ | Jacob (2000) |
| $k_{BR,4}$ | Rate constant, R8 | $1.3 \times 10^{-11}$ cm$^3$ s$^{-1}$ | Jacob (2000) |
| $k_{BR,5}$ | Rate constant, R9 | $2.5 \times 10^{-12}$ cm$^3$ s$^{-1}$ | Jacob (2000) |
| $k_{BR,6}$ | Rate constant, R10 | $1.6 \times 10^{-11}$ cm$^3$ s$^{-1}$ | Jacob (2000) |
| $k_{GP}$ | Rate constant, R3 | $3 \times 10^{-12}$ cm$^3$ s$^{-1}$ | Sander et al. (2003) |
| $K_{eq}$ | Equilibrium constant, R4 | $2.1 \times 10^{-5}$ M | Thornton et al. (2008) |
| $H_{HO2}$ | $HO_2$ Henry's law constant | 5600 M atm$^{-1}$ | Thornton et al. (2008) |
| $\tau_d$ | $HO_2$ desorption lifetime | $1.5 \times 10^{-3}$ s | Shiraiwa et al. (2010) |
| $\alpha_{s,0}$ | $HO_2$ surface accommodation at time 0 | 0.22 | |
| $D_{g,HO2}$ | $HO_2$ gas phase diffusion rate constant | 0.25 cm$^{-2}$ s$^{-1}$ | Thornton et al. (2008) |
| $[Cu]$ | Copper concentration (used when modelling copper doped sucrose aerosol particles) | $5 \times 10^{19}$ cm$^{-3}$ | |
| $T$ | Temperature | 293 K | |









**Table 2:** Summary of the reactants and conditions that were utilised and the HO$_2$ uptake coefficients
that were measured during the experiments. Experiments 1 - 4 were performed using the smog
chamber whereas experiments 5 - 9 utilised the PAM chamber.

| Experiment number | Reaction type | Initial precursor concentrations | UV | Relative humidity in the chamber/ % | Pressure in the flow tube/ mbar | Maximum aerosol surface to volume ratio in the flow tube/ cm$^2$ cm$^{-3}$ | HO$_2$ uptake coefficient ($\gamma$) |
|---|---|---|---|---|---|---|---|
| 1 | α-pinene ozonolysis | [α-pinene] = 600 ppb [O$_3$] = 280 ppb | Off | 50 | 987 | $6.30 \times 10^{-5}$ | < 0.01 |
| 2 | α-pinene ozonolysis | [α-pinene] = 600 ppb [O$_3$] = 280 ppb | Off | 50 | 965 | $1.30 \times 10^{-4}$ | < 0.004 |
| 3 | α-pinene ozonolysis | [α-pinene] = 200 ppb [O$_3$] = 310 ppb | Off | 80 | 939 | $7.10 \times 10^{-5}$ | < 0.006 |
| 4 | α-pinene photochemistry | [α-pinene] = 500 ppb [NO$_2$] = 350 ppb | On | 50 | 940 | $6.30 \times 10^{-5}$ | < 0.018 |
| 5 | α-pinene photochemistry | [α-pinene] = 500 ppb | On | 50 | 929 | $2.93 \times 10^{-4}$ | < 0.001 |
| 6 | TMB photochemistry | [TMB] = 2 ppm | On | 50 | 923 | $2.75 \times 10^{-4}$ | 0.004 ± 0.002 |
| 7 | TMB photochemistry | [TMB] = 2ppm | On | 50 | 918 | $2.32 \times 10^{-4}$ | 0.004 ± 0.003 |
| 8 | α-pinene photochemistry | [α-pinene] = 500 ppb | On | 50 | 927 | $1.88 \times 10^{-4}$ | < 0.005 |
| 9 | α-pinene photochemistry | [α-pinene] = 1 ppm | On | 80 | 904 | $3.90 \times 10^{-4}$ | < 0.001 |

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
