# Peer review of "The effect of viscosity and diffusion on the HO2 uptake by sucrose and secondary organic aerosol particles"

_Atmospheric Chemistry and Physics, 2016_

## Referee Comment (RC1) · Anonymous Referee #2 · 22 May 2016

This manuscript provides an important step towards understanding the uptake of HO2 by secondary organic aerosol (SOA) and the influence of the viscous/glassy nature of the SOA on the reactive uptake kinetics. The manuscript is well-written, concise, and the data are clearly presented. I have only minor suggestions for the authors to consider before the manuscript can be accepted for publication.

Line 269: "There is good agreement between the model and the measurements suggesting that the change in HO2 uptake over the range of humidities is indeed due to a change in the HO2 diffusion coefficient which is in turn due to a change in the viscosity of the aerosol particles." Given that the viscosity of aqueous/sucrose aerosol changes by more than >8 orders of magnitude over this experimental RH range (10 Pa s to >109 Pa s), things are clearly more complex than this sentence suggests with uptake coefficients only changing by a little over 1 order of magnitude. The authors do go on to

discuss this more fully and they seem to attributed it to a combination of increasing viscosity, which suppresses diffusion rates into the particle bulk, coupled with an efficient surface reaction. In effect, the surface reaction means the uptake coefficient doesn't decreases as much as it might based on the increase in viscosity (see the dashed and solid lines in Figure 5). The near-surface chemistry is attributed to the presence of catalysing Cu2+ ions near the surface.

It would be helpful to the reader to compare directly the relative change in viscosity expected over this range to the change in the uptake coefficient (see for example the recent paper by Marshall et al., Diffusion and Reactivity in Ultraviscous Aerosol and the Correlation with Particle Viscosity, Chem. Sci., 7, 1298–1308, doi:10.1039/C5SC03223G, 2016 where they do something similar for previous measurements). Indeed, many of the studies referenced in the introduction have shown that the uptake coefficients for OH, O3 and N2O5 etc do not vary nearly as strongly with viscosity as would be expected and do infact only decrease by an order of magnitude, even though the viscosity/diffusivity changes by many more orders of magnitude.

In these earlier studies, it is not clear that there is any special surface enhanced chemistry that keeps the uptake coefficient larger than would be expected. Further clarification/discussion is needed. How can the authors be sure that that the inclusion of a surface reaction is what is needed to "correct" for a diminishing bulk diffusion inferred from water diffusion constants and does anything other than provide an additional parameter (degree of freedom) with which to ensure good agreement between the measurements and model? How legitimate do the authors believe it is to use of the diffusion constant for water (a stable molecule forming 2 hydrogen bonds etc) to represent the much more reactive, less strongly interacting HO2 in viscous sucrose? In the other measurements of reactive uptake coefficients, must similar enhancements in surface reaction rates be included to explain why the uptake coefficients do not fall as much as would be expected based on viscosity and water diffusivity?

Line 293: "Bones et al. (2012) measured that for 100 nm diameter sucrose aerosol

particles..." – to my knowledge, this paper does not report measurements of these timescales for size change for such small particles, it only inferred them from measurements on larger particles.

Line 309: The two bulk concentrations of copper sulfate were chosen to span the expected range based on RH. Does this mean the concentration is expected to be 0.1 M at 17 % RH? How might the pH be expected to vary for supersaturated solutions of sucrose containing copper sulfate at such low water activity?

Line 366: "...effect of aerosol viscosity upon $HO_2$ uptake coefficients was systematically investigated with a combination of $HO_2$ uptake coefficient measurements..." – as suggested, it might be helpful to show the dependence of uptake coefficient on viscosity explicitly.

Minor Points for Clarification/Additional Information

Line 188: What is the timescale for $HO_2$ concentration to stabilise once mercury lamp turned on?

Line 210: Magnitude of $RO_2$ interference signal in $HO_2$ detection is shown to be significant for TMB SOA measurements but not a-pinene – this is different from expectations based on box model simulations. Why is this the case? Some more detailed discussion would be helpful.

Line 219: Discussion of correction for wall loss and non-plug flow would benefit from indicating directly the level of correction typically required beyond what can be inferred from Figure 3.

Line 238: "which is only slightly larger than the diameter of $HO_2$ (0.4 nm)." – is there any significance to this?

---

## Referee Comment (RC2) · Anonymous Referee #1 · 24 May 2016

The manuscript entitled, "The effect of viscosity on the HO2 uptake by sucrose and secondary organic aerosol particles," by Lakey et. al. describes new measurement of the uptake coefficient of HO2 onto model organic aerosol surfaces (sucrose and SOA). The HO2 loss rate onto aerosol surfaces may have important ramifications for the HOx cycle in the atmosphere. The experimental measurements are interpreted using a kinetic multi-layer model. The experimental measurements are carefully done with the appropriate checks and associated error analysis. The following aspects should be addressed by the authors in a revised manuscript.

1. There are a number of studies not cited in the manuscript that have generally discussed and observed the evolution of uptake coefficients with particle viscosity. These should be included and discussed in light of their relevance for the present study. For example, Slade and Knopf (Geophys. Res. Lett., 41, 5297–5306), Davies and Wilson

(Chem Sci, 2015,6, 7020-7027) Houle et al. (Phys. Chem. Chem. Phys., 2015, 17, 4412), observed similar relationships as shown in Fig. 5 albeit for OH radical uptake. All of these studies interpret the relationship between gamma and viscosity mathematically using diffusoreactive lengths and such an analysis should be conducted in order to connect the present study with the larger body of literature on the subject.

2. On lines 275-279 the authors say that the Henry's law constant and HO2 diffusion constant are both terms contained within the bulk HO2 bulk accommodation coefficient. This statement should be clarified in the manuscript. Does this mean that the two quantities cannot be separated independently in the model?

3. The authors should clarify the geometry of the simulation, spherical shells?

4. As diffusion becomes slower as RH decreases for sucrose, will any of the rate coefficients for R4-R10 be limited by this slow diffusion. In other words, while diffusion of species between shells decreases with RH, do the reactions occurring within a single shell have to be slowed due to diffusion? This should be clarified why this is or is not needed to accurately simulate the experimental data.

5. The desorption lifetime of HO2 appears unphysically long (1.5 ms) compared to MD simulations results presented in Vieceli et al. (J. Phys. Chem. B 2005, 109, 15876-15892). The desorption lifetimes from MD for OH, O3 and H2O range from 35-140 ps, orders of magnitude faster than the HO2 value used in the present study. While these are different species, I can't imagine a molecular reason for such a large difference for HO2. The value used here for the desorption lifetime of HO2 (Shiraiwa, 2010) is from a previous simulation study of ozone + oleic acid and appears to be a fit parameter, which is also orders of magnitude different than the MD results of Vieceli et al. for the ozone desorption lifetime. The reason for this discrepancy should be addressed and if possible simulations be run with a more realistic value for t_d.

6. To account for the finite value of the gamma at low RH the authors invoke a surface reaction (R11) catalyzed by Cu+2. It is unclear what the rate law is for this catalyzed

reaction, the authors should make this clear in the manuscript. Furthermore, why would this reaction only occur at the surface? This assumes that HO2, which is fairly unreactive will not diffuse even into the subsurface region. It seems more reasonable to me, since surface rate coefficients are rarely measured, to allow this reaction to occur in the subsurface with a rate coefficient that has bulk units. This will allow the reader to more closely evaluate the magnitude of the rate coefficient needed to account for the measurements.

7. The authors interpret their data solely in terms of the diffusion of HO2 into sucrose. However, it is not clear whether this captures the complete picture. Wouldn't the diffusion of Cu+2, and O2- to the surface be equally important as HO2 diffusion at low RH for determining gamma? The authors should include these diffusing species in the simulation or clearly justify why it is not necessary to explicitly include the diffusion of all species (including sucrose) in their simulation.

---

## Author Comment (AC1) · 18 Aug 2016

The comment was uploaded in the form of a supplement:
http://www.atmos-chem-phys-discuss.net/acp-2016-284/acp-2016-284-AC1-supplement.pdf

---

## Author Response (AR1)

We thank the two reviewers for their very helpful comments, which have helped to improve the manuscript. The detailed point-by-point responses to the referees' comments are given below in blue and we have indicated the changes made in the manuscript in response to each comment.

Anonymous Referee #1

The manuscript entitled, "The effect of viscosity on the HO2 uptake by sucrose and secondary organic aerosol particles," by Lakey et. al. describes new measurement of the uptake coefficient of HO2 onto model organic aerosol surfaces (sucrose and SOA). The HO2 loss rate onto aerosol surfaces may have important ramifications for the HO$_x$ cycle in the atmosphere. The experimental measurements are interpreted using a kinetic multi-layer model. The experimental measurements are carefully done with the appropriate checks and associated error analysis. The following aspects should be addressed by the authors in a revised manuscript.

1.  There are a number of studies not cited in the manuscript that have generally discussed and observed the evolution of uptake coefficients with particle viscosity. These should be included and discussed in light of their relevance for the present study. For example, Slade and Knopf (Geophys. Res. Lett., 41, 5297–5306), Davies and Wilson (Chem Sci, 2015,6, 7020-7027) Houle et al. (Phys. Chem. Chem. Phys., 2015, 17, 4412), observed similar relationships as shown in Fig. 5 albeit for OH radical uptake. All of these studies interpret the relationship between gamma and viscosity mathematically using diffusoreactive lengths and such an analysis should be conducted in order to connect the present study with the larger body of literature on the subject.

    We have now included these references within the manuscript. We have calculated the reacto-diffusive length as varying between ~ 4 – 7 nm at the highest relative humidity that we used (71 % RH) down to ~0.006 – 0.05 nm at the lowest relative humidity that we used (17 % RH). The range of values for the reacto-diffusive length is due to the difference of the parameterizations of the diffusion coefficient given by (Price *et al.*, 2014) and (Zobrist *et al.*, 2011). These reacto-diffusive lengths indicate that at all relative humidities HO$_2$ radicals are limited to the outermost molecular layers of the particle, which is in agreement with our model. In these previous studies, it was shown that as the reacto-diffusive length increased, the uptake of gas-phase species would also increase as the gas-phase species could travel further into the bulk of the aerosol. Similarly, in our work, as the reacto-diffusive length increases and the HO$_2$ can travel further into the aerosol bulk an increase in the HO$_2$ uptake coefficient is observed until the uptake coefficient is limited by surface mass accommodation. The effect of diffusion and the reacto-diffusive length is large, as can be observed by the dashed-lines in Figure 5. However, at the lower relative humidities where the reacto-diffusive length is significantly smaller than the diameter of HO$_2$ a surface reaction is required for the uptake coefficient to not be underestimated. It should also be noted that in contrast to the case of OH, the slower and more complex chemistry of HO$_2$ requires explicit modelling and cannot be treated with a simplified resistor model.

    This discussion has been added to the results section of the manuscript with the added text shown below:
    *"The HO$_2$ reacto-diffusive length (Schwartz and Freiberg, 1981, Hanson et al., 1994) varied from between ~ 4 – 7 nm at the highest relative humidity that was used (71 % RH) down to ~0.006 – 0.05 nm at the lowest relative humidity (17 % RH). The range of values for the reacto-diffusive length at a given RH is due to the difference between the parameterizations of the diffusion coefficient in Price et al. (2014) and Zobrist et al. (2011). These reacto-diffusive lengths indicate that at all relative humidities HO$_2$ radicals will be limited to the outermost molecular layers of the particle before reacting away, which is in agreement with the model. Note that it was shown in previously that the uptake of gas-phase species generally increases with increasing reacto-diffusive length,*

*which is consistent with our HO₂ uptake coefficient measurements (Slade and Knopf, 2014, Davies and Wilson, 2015, Houle et al., 2015)."*

*"In contrast to traditional resistor models, the KM-SUB model enables efficient treatment of complex chemical mechanisms."*

2. On lines 275-279 the authors say that the Henry's law constant and HO2 diffusion constant are both terms contained within the bulk HO2 bulk accommodation coefficient. This statement should be clarified in the manuscript. Does this mean that the two quantities cannot be separated independently in the model?

We have now changed bulk accommodation to mass accommodation within the manuscript. Mass accommodation, which is the mass transfer from the gas phase to the bulk of the aerosol, contains two different sub-processes: surface accommodation and surface-to-bulk transfer. The surface-to-bulk transfer rate describes transport from the surface of the aerosol particle into bulk layer 1 and is a derived input parameter in our model. It is dependent upon both the Henry's law constant and the HO₂ diffusion coefficient (Pöschl *et al.*, 2007, Berkemeier *et al.*, 2013). These two quantities can still be independently changed within the model and would both affect the HO₂ uptake coefficient. It makes sense that the rate at which a molecule is incorporated into the bulk of a solution is inversely dependent on viscosity ((Behr *et al.*, 2009), and references therein), similarly to the diffusivity being inversely dependent upon the viscosity as stated by the Stokes Einstein equation. Therefore, the Henry's law constant and HO₂ diffusion coefficient are coupled within the process of mass accommodation.

The text has been reworded to show that the Henry's law constant and HO₂ diffusion coefficient can be changed independently. The clarified text is shown below:
*"At intermediate relative humidities with γ < α_{s,0}, the uptake is limited by surface-to-bulk transport, which is related to both solubility (Henry's law coefficient) and diffusivity (diffusion coefficient) in the kinetic model. Under both conditions, the uptake is driven by chemical reaction in the near-surface bulk and effectively limited by mass accommodation, which includes both surface accommodation and surface-to-bulk transport (Behr et al., 2009, Berkemeier et al., 2013)."*

3. The authors should clarify the geometry of the simulation, spherical shells?
Yes, the shells used within the model were spherical. This has now been clarified within the manuscript by modifying the following sentence:
*"It is a multi-layer model comprising a gas phase, a near-surface gas phase, a sorption layer, a near-surface bulk layer and a number of bulk layers arranged in spherical geometry."*

4. As diffusion becomes slower as RH decreases for sucrose, will any of the rate coefficients for R4-R10 be limited by this slow diffusion. In other words, while diffusion of species between shells decreases with RH, do the reactions occurring within a single shell have to be slowed due to diffusion? This should be clarified why this is or is not needed to accurately simulate the experimental data.

The rate coefficients for reactions R4-R10 were assumed constant over the entire range of relative humidities. However, it is expected that if the reactions are diffusion limited, decreasing the diffusion coefficients would decrease the collision rates between molecules and thus the rate coefficients would also decrease. Currently, our model predicts that at low relative humidity the uptake is limited by mass accommodation due to slow surface-to-bulk transfer and at high relative humidity it is limited by surface mass accommodation. We therefore performed sensitivity tests to check whether a decrease in the rate constants would affect the outputted uptake coefficient. We found that when decreasing all reaction rate constants by a factor of 100, the largest effect was

observed when the $HO_2$ diffusion coefficients were in the range of $10^{-10} – 10^{-9}$ (equivalent to a RH of ~ 40 – 55 %) with a decrease in the uptake coefficient of slightly less than 10 %. The model predicts that for this range of diffusion coefficients, when decreasing the rate constants by a factor of 100, the $HO_2$ uptake goes from being mass accommodation limited to bulk reaction limited. At the lower relative humidities the uptake would still be mass accommodation and surface reaction limited and thus independent of the bulk reaction rate, whereas at the higher relative humidities the uptake coefficient is still limited by surface mass accommodation. It should also be noted that in the hypothetical case of no surface reaction, if we were to further decrease the reaction rates at the lower relative humidities (< 35 % RH) such that the uptake was limited by bulk reaction rather than mass accommodation, the difference between the measurements and the model output would become larger. This indicates the importance of the surface reaction.

The caption of Table 1 has been updated to show that the parameters shown in Table 1 are for all relative humidities. The updated caption is shown below:
*"The parameters used in the KM-SUB $HO_2$ uptake model over all relative humidities."*
The following text has also been added to show the impact of decreasing the rate constants by a factor of 100.
*"A two order of magnitude decrease in the rate constants affects the uptake coefficient marginally by reducing it by less than 10 % in the 40 – 55 % relative humidity range, but has no impact at the lower or higher relative humidities."*

5.  The desorption lifetime of HO2 appears unphysically long (1.5 ms) compared to MD simulations results presented in Vieceli et al. (J. Phys. Chem. B 2005, 109, 15876-15892). The desorption lifetimes from MD for OH, O3 and H2O range from 35-140 ps, orders of magnitude faster than the HO2 value used in the present study. While these are different species, I can't imagine a molecular reason for such a large difference for HO2. The value used here for the desorption lifetime of HO2 (Shiraiwa, 2010) is from a previous simulation study of ozone + oleic acid and appears to be a fit parameter, which is also orders of magnitude different than the MD results of Vieceli et al. for the ozone desorption lifetime. The reason for this discrepancy should be addressed and if possible simulations be run with a more realistic value for t_d.

Much discussion was spent in the past on desorption lifetimes of reactive molecules in kinetic flux models. In early studies (Pfrang *et al.*, 2010, Shiraiwa *et al.*, 2010), desorption lifetimes in the millisecond range were used, a value determined by manual fitting. In follow-up studies, much shorter desorption lifetimes of ozone physisorption were used to be compliant with the results from MD simulations (Shiraiwa *et al.*, 2011). However, in this study, the effective residence time of atmospheric oxidants on aerosol surfaces also remained high through inclusion of long-lived reactive oxygen intermediates. Berkemeier *et al.* (2013) showed that when using desorption lifetimes in the nanosecond range, not including reactive oxygen intermediates, surface reaction is way too slow to reach observed uptake coefficients in heterogeneous reaction systems and never influences reactivity: A desorption lifetime in that range effectively disables Langmuir-Hinshelwood type surface reactions as surface concentrations are too low to have impact on total reactivity with reasonable surface reaction rate coefficients. This stands in contrast to several studies showing the importance of surface reactions on aerosols. Nanosecond desorption lifetimes hence also effectively disable the possibility of surface saturation effects commonly observed in aerosol kinetics. Steimer *et al.* (2015a) show a strong gas phase concentration dependence of ozone uptake onto shikimic acid films with a Langmuir constant of ~$6 \times 10^{-13}$ $cm^3$. Such a high Langmuir constant cannot be obtained with a nanosecond desorption lifetime. The surface adsorption of reactive trace gases on aerosol surfaces must thus be a process that goes beyond simple physisorption. Possibly, aerosol surface chemistry has to include chemical reaction occurring after partial dissolution of reactants in the uppermost layer of organics, like discussed in the case of oleic acid ozonolysis in

Hearn *et al.* (2005), or be some kind of chemisorption process as discussed in Shiraiwa *et al.* (2011) and Berkemeier *et al.* (2016). MD simulations look at mere physically adsorbed, and hence weakly bound, molecules desorbing from a flat surface. We are not sure about the process extending the effective desorption lifetime of atmospheric trace species and hence $HO_2$ in our model, but we are confident that the effective desorption lifetimes to be used in models employing a in Langmuir-Hinshelwood-type surface reaction scheme must lie in the range of at least several microseconds to describe all the surface effects that can be observed in laboratory experiments. Uptake of $HO_2$ onto mineral dust particles (where only surface reaction is possible) would not be significant if $HO_2$ molecules would only reside on the surface for a few nanoseconds. We will follow-up on this issue in future publications and added the following clarifying comment to the manuscript:

*"Note however that for a relevant surface reaction in kinetic flux models, it is necessary to use an effective desorption lifetime $\tau_d$ in the millisecond to second time range (Shiraiwa et al., 2010, Berkemeier et al., 2016). This is many orders of magnitude longer than would be expected due to pure physisorption as estimated by molecular dynamic simulations (Vieceli et al., 2005), indicating that the adsorption process should involve chemisorption or formation of long-lived intermediates that would have the potential to extend these effective desorption lifetimes (Shiraiwa et al., 2011, Berkemeier et al., 2016)."*

6. To account for the finite value of the gamma at low RH the authors invoke a surface reaction (R11) catalyzed by Cu+2. It is unclear what the rate law is for this catalyzed reaction, the authors should make this clear in the manuscript. Furthermore, why would this reaction only occur at the surface? This assumes that HO2, which is fairly unreactive will not diffuse even into the subsurface region. It seems more reasonable to me, since surface rate coefficients are rarely measured, to allow this reaction to occur in the subsurface with a rate coefficient that has bulk units. This will allow the reader to more closely evaluate the magnitude of the rate coefficient needed to account for the measurements.

The surface reaction was required to describe the experimental data at the lower relative humidities due to it being independent of the bulk diffusion coefficients. For this surface reaction we assume a bimolecular self-reaction, $HO_2 + HO_2$. The actual reaction mechanism remains unclear and could potentially be $Cu + HO_2$ or an even more complex mechanism. However, the model results are insensitive to the true mechanism occurring at the surface of the aerosol particle which is thus beyond the scope of this work. We have used a simple mechanism to demonstrate that a surface reaction must be occurring. By multiplying the diameter of $HO_2$ (0.4 nm) by the surface rate constant ($1 \times 10^{-8}$ $cm^2$ $s^{-1}$) we can estimate an equivalent bulk reaction rate constant of ~ $4 \times 10^{-16}$ $cm^3$ $s^{-1}$ (which is equivalent to $2.4 \times 10^5$ $M^{-1}$ $s^{-1}$).

In the particle bulk however, a Cu-catalyzed reaction (reactions R7 and R9) is explicitly treated in addition to the bimolecular self-reaction of $HO_2$ (reaction R5). $HO_2$ diffusion into the sub-surface (or bulk layer 1 within our model) is included within the model and may exhibit the mass accommodation-limited uptake discussed in the manuscript (Berkemeier *et al.*, 2013). It should be noted that the reaction rate coefficients within bulk layer 1 would be the same as in all other bulk layers. Mass accommodation will still occur at the lower relative humidities but to a much lesser extent due to the lower diffusion coefficients.

We have specified within the text that the reaction included at the surface is the $HO_2$ self-reaction but that the mechanism remains unclear. The following text has been slightly modified to include these points:

*"For example, at 17 % RH and without a surface reaction, $\gamma$ values as low as $\sim 5 \times 10^{-4}$ and $\sim 3 \times 10^{-5}$ would be expected using the Zobrist et al. (2011) and Price et al. (2014) parameterisations, respectively. However, by including the following self-reaction of $HO_2$ at the surface of the sucrose particles, much better agreement with the observed values of around $\sim 10^{-2}$ could be obtained (Fig. 5)."*

*"Although the true mechanism for reaction at the surface remains unclear, the large rate constant for this reaction suggests that copper could potentially be catalyzing the destruction of $HO_2$ at the surface of the sucrose particles which is consistent with the higher $HO_2$ uptake coefficients measured onto solid aerosol particles containing transition metals compared to solid aerosol particles containing no transition metal ions (Bedjanian et al., 2013, George et al., 2013, Matthews et al., 2014, Lakey et al., 2015)."*

7.  The authors interpret their data solely in terms of the diffusion of HO2 into sucrose. However, it is not clear whether this captures the complete picture. Wouldn't the diffusion of Cu+2, and O2- to the surface be equally important as HO2 diffusion at low RH for determining gamma? The authors should include these diffusing species in the simulation or clearly justify why it is not necessary to explicitly include the diffusion of all species (including sucrose) in their simulation.

    We explicitly treat diffusion of $O_2^-$ and $Cu^{2+}$ in the model and use the same diffusion coefficients for these species as for $HO_2$. However, sensitivity studies with independent diffusion coefficients showed that only changing the diffusion rate of $HO_2$ affected modelling results. A range of values were tested, however, the $HO_2$ uptake coefficient outputted by the model did not change when setting the diffusion coefficient of $O_2^-$, $Cu^{2+}$ and $Cu^+$ to zero or changing it to be 1000 times faster than the diffusion coefficients of $HO_2$ and over the whole range of relative humidities. For $O_2^-$ the diffusion coefficient is unimportant due to the rates of reaction with copper being so rapid that it is produced in situ and consumed locally. The diffusion of $Cu^+$ and $Cu^{2+}$ would only be important within the model if the reaction with $HO_2/ O_2^-$ permanently removed them or if their rate coefficients with $HO_2$ were significantly different such that the equilibrium between the two ions was important. However, as the reactions with $HO_2/ O_2^-$ are catalytic and cause $Cu^{2+}$ and $Cu^+$ to rapidly interconvert they are still available at high concentrations in the upper layers of the aerosol particle. Similarly, as sucrose does not react with any species in the model, its diffusion within the model is unimportant to the outputted $HO_2$ uptake coefficient.

    This discussion has been added to the model description section as shown below.
    *"Sensitivity tests showed that the diffusion rate constants of $O_2^-$, $Cu^+$ and $Cu^{2+}$ did not influence calculation results. The reaction rate coefficients involving copper ($k_{BR,3}$ - $k_{BR,6}$) are so large that $O_2^-$ is produced in situ and consumed locally. The catalytic nature of these reactions cause $Cu^+$ and $Cu^{2+}$ to rapidly interconvert meaning that they remain available at high concentrations in the upper layers of the aerosol particle. Similarly, as sucrose does not react with any species within the model, its diffusion within the model is unimportant to the outputted $HO_2$ uptake coefficient."*

Anonymous Referee #2

This manuscript provides an important step towards understanding the uptake of HO2 by secondary organic aerosol (SOA) and the influence of the viscous/glassy nature of the SOA on the reactive uptake kinetics. The manuscript is well-written, concise, and the data are clearly presented. I have only minor suggestions for the authors to consider before the manuscript can be accepted for publication.

Line 269: "There is good agreement between the model and the measurements suggesting that the change in HO2 uptake over the range of humidities is indeed due to a change in the HO2 diffusion

coefficient which is in turn due to a change in the viscosity of the aerosol particles." Given that the viscosity of aqueous/sucrose aerosol changes by more than >8 orders of magnitude over this experimental RH range (10 Pa s to >109 Pa s), things are clearly more complex than this sentence suggests with uptake coefficients only changing by a little over 1 order of magnitude. The authors do go on to discuss this more fully and they seem to attributed it to a combination of increasing viscosity, which suppresses diffusion rates into the particle bulk, coupled with an efficient surface reaction. In effect, the surface reaction means the uptake coefficient doesn't decrease as much as it might based on the increase in viscosity (see the dashed and solid lines in Figure 5). The near-surface chemistry is attributed to the presence of catalysing Cu2+ ions near the surface. It would be helpful to the reader to compare directly the relative change in viscosity expected over this range to the change in the uptake coefficient (see for example the recent paper by Marshall et al., Diffusion and Reactivity in Ultraviscous Aerosol and the Correlation with Particle Viscosity, Chem. Sci., 7, 1298–1308, doi:10.1039/C5SC03223G, 2016 where they do something similar for previous measurements).

Although the viscosity changes by more than 8 orders of magnitude over this range of relative humidities the diffusion coefficients only change by ~ 5 – 7 orders of magnitude depending on whether the Zobrist *et al.* (2011) or Price *et al.* (2014) parameterizations are used. Uptake coefficients are dependent upon the square root of the diffusion coefficient (Davidovits *et al.*, 2006). Therefore, we expect (based purely on diffusion), that the uptake coefficients would change by 2.5 – 3.5 orders of magnitude over the range of relative humidities. However, measured $HO_2$ uptake coefficients change by ~ 1 order of magnitude due to the surface mass accommodation limiting the uptake coefficient at the higher relative humidities and a surface reaction becoming important at the lower relative humidities. In order to demonstrate how the uptake coefficient changes as a function of viscosity we have now added an extra panel to Figure 5 (see the modified figure and new caption after another response below).

Indeed, many of the studies referenced in the introduction have shown that the uptake coefficients for OH, O3 and N2O5 etc do not vary nearly as strongly with viscosity as would be expected and do infact only decrease by an order of magnitude, even though the viscosity/diffusivity changes by many more orders of magnitude. In these earlier studies, it is not clear that there is any special surface enhanced chemistry that keeps the uptake coefficient larger than would be expected. Further clarification/discussion is needed. How can the authors be sure that that the inclusion of a surface reaction is what is needed to "correct" for a diminishing bulk diffusion inferred from water diffusion constants and does anything other than provide an additional parameter (degree of freedom) with which to ensure good agreement between the measurements and model? How legitimate do the authors believe it is to use of the diffusion constant for water (a stable molecule forming 2 hydrogen bonds etc) to represent the much more reactive, less strongly interacting HO2 in viscous sucrose? In the other measurements of reactive uptake coefficients, must similar enhancements in surface reaction rates be included to explain why the uptake coefficients do not fall as much as would be expected based on viscosity and water diffusivity?

Sensitivity tests showed that at the lower relative humidities the $HO_2$ uptake coefficient was only limited by three parameters: the $HO_2$ Henry's law constant, the $HO_2$ diffusion coefficient and the rate coefficient of the surface reaction. As already discussed in the manuscript the effective Henry's law constant is very sensitive to the pH of the aerosol due to the equilibrium between $HO_2$ and $O_2^-$ (R4). The pH of the aerosols would have to increase significantly to almost neutral pH values and greater than 7 for the parameterization of (Zobrist *et al.*, 2011) and (Price *et al.*, 2014), respectively, in order to obtain $HO_2$ uptake coefficients that could reproduce the measured $HO_2$ uptake coefficient at the lowest relative humidity. It seems extremely unlikely that the pH would increase significantly at the lower relative humidities as pH measurements of the expected copper sulfate concentrations within the aerosol particles showed that the pH should be ~ 4.1. It should also be noted that in case of a higher pH, the measurements at the intermediate relative humidities could no longer be reproduced by the model. We

have also not found any evidence in the literature for salting in of $HO_2$ as the organic fraction increases at low relative humidity, but if this does occur the effect is likely to be minimal and would change the uptake coefficient by less than 1 order of magnitude. Hydrogen bonding of water with sucrose could reduce the diffusion coefficients of water through a sucrose particle whilst $HO_2$, which is less strongly interacting, would have comparatively larger diffusion coefficients. However, it is also possible that hydrogen bonding could aid the diffusion process by smoothing out diffusion barriers. However, we see no evidence of this happening at the higher relative humidities (RH > 40%) where the $HO_2$ uptake coefficients can be reproduced well using $HO_2$ diffusion coefficients based upon the diffusion coefficients of water. The shape of the data in Figure 5 suggests a surface reaction as the decrease in the uptake coefficient at the low relative humidities is much smaller than would be expected from a constant decrease in the diffusion coefficients. It should also be noted that the effect and importance of surface reactions is consistent with previous work by Gržinić *et al.* (2015) and Berkemeier *et al.* (2016) for the uptake of $N_2O_5$ to citric acid and the uptake of $O_3$ to shikimic acid over a range of relative humidities. For these examples the surface reaction was rate limiting under certain conditions so that its extent could be better quantified than in the present case.

We have clarified within the results section that the inclusion of a surface reaction is consistent with previous studies by adding the following text:

*"The effect and importance of surface reactions is consistent with previous work by Gržinić et al. (2015), Steimer et al. (2015b) and Berkemeier et al. (2016) for the uptake of $N_2O_5$ to citric acid and the uptake of $O_3$ to shikimic acid over a range of relative humidities."*

We have also emphasized that the uptake coefficient varies less than might be expected based on the large change in viscosity:

*"Although the viscosity changes by more than 8 orders of magnitude and the diffusion coefficients change by 5-7 orders of magnitude over the investigated range of relative humidity, the measured $HO_2$ uptake coefficients change by only ~ 1 order of magnitude. This can be explained to some extent by the uptake coefficient being proportional to the square root of the diffusion coefficient when the uptake is controlled by reaction and diffusion of $HO_2$ in the bulk (Davidovits et al., 2006, Berkemeier et al., 2013). If this were the only mechanism involved, however, one would still expect a change in the uptake coefficient by 2.5 – 3.5 orders of magnitude."*

We have also added the following Figure to the manuscript in order to demonstrate the limiting cases within the model at different relative humidities.

[Figure]

*Figure 6: The kinetic cube representing the eight limiting cases for uptake of gases to aerosol particles (Berkemeier et al., 2013). $B_{rx}$: bulk reaction limited by chemical reaction, $B_{bd}$: bulk reaction limited by bulk diffusion of the volatile reactant and the condensed reactant, $B_a$: bulk reaction limited by mass accommodation, $B_{gd}$: bulk reaction limited by gas-phase diffusion; $S_{rx}$: surface reaction limited by chemical reaction, $S_{bd}$: surface reaction limited by bulk diffusion of a condensed reactant, $S_a$: surface reaction limited by surface accommodation, $S_{gd}$: surface reaction limited by gas-phase diffusion. For copper doped sucrose aerosol particles, the $HO_2$ uptake coefficient is limited by mass accommodation under humid conditions and by chemical reaction at the surface at low relative humidity.*

Line 293: "Bones et al. (2012) measured that for 100 nm diameter sucrose aerosol
Particles to my knowledge, this paper does not report measurements of these timescales for size change
for such small particles, it only inferred them from measurements on larger particles.
We have clarified within the text that this was inferred from measurements on larger particles as shown
below:
*"Bones et al. (2012) inferred from measurements on larger particles that for 100 nm diameter sucrose
aerosol particles the equilibration time would be more than 10 seconds when the viscosity increased
above ~ $10^5$ Pa s, which would occur at ~ 43 % RH (Power et al., 2013)."*

Line 309: The two bulk concentrations of copper sulfate were chosen to span the expected range based
on RH. Does this mean the concentration is expected to be 0.1 M at 17 % RH? How might the pH be
expected to vary for supersaturated solutions of sucrose containing copper sulfate at such low water
activity?

Yes, our calculations predicted a concentration of copper of 0.1 M at 17 % RH. Unfortunately, we cannot
directly measure the aerosol pH within the supersaturated solutions of sucrose. However, as sucrose

within water has a pH of 7 and a very high pKa of 12.6, we expect the pH to be dominated by the presence of copper sulfate within the aerosol particles.

The following text has been added to the manuscript:

"*It is expected that the pH would be dominated by the presence of copper sulfate rather than sucrose which has a pH of 7 in water and a very high pKa of 12.6.*"

Line 366: "effect of aerosol viscosity upon HO2 uptake coefficients was systematically investigated with a combination of HO2 uptake coefficient measurements" as suggested, it might be helpful to show the dependence of uptake coefficient on viscosity explicitly.

We agree that it is interesting to show the dependence of the uptake coefficient upon the viscosity explicitly and we have now added an extra panel to Figure 5 as shown below based upon the data and fitting shown in Power *et al.* (2013) and Marshall *et al.* (2016):

[Figure]

**Figure 5:** The $HO_2$ uptake coefficient onto copper (II) doped sucrose aerosol particles as a function of (a) relative humidity and (b) aerosol particle viscosity. The lines represent the expected $HO_2$ uptake coefficient calculated using the KM-SUB model using the Price et al. (2014) (red) and Zobrist et al. (2011) (blue) diffusion parameterisations (see model description section) and with (solid) and without (dashed) the inclusion of a surface reaction (Reaction R11). The viscosity within sucrose aerosol particles is based upon the data and fitting shown in Power et al. (2013) and Marshall et al. (2016) whilst the red and blue axes in panel (a) are the Price et al. (2014) and Zobrist et al. (2011) diffusion parameterisations, respectively. The error bars represent two standard deviations of the propagated error in the gradient of the $k'$ against aerosol surface area graphs.

Minor Points for Clarification/Additional Information

Line 188: What is the timescale for HO2 concentration to stabilise once mercury lamp turned on?

The $HO_2$ concentration within the flow tube will stabilize in less than 1 minute when the mercury lamp is switched on. We always wait for the $HO_2$ concentration to be stable before starting a $HO_2$ decay.

This has been clarified within the manuscript by adding the following text:

*"Data acquisition was only started once HO$_2$ concentrations within the flow tube were stable which occurred within 1 minute of switching on the mercury lamp."*

Line 210: Magnitude of RO2 interference signal in HO2 detection is shown to be significant for TMB SOA measurements but not a-pinene – this is different from expectations based on box model simulations. Why is this the case? Some more detailed discussion would be helpful.

As stated in the manuscript, the expected interference from TMB RO$_2$ and α-pinene RO$_2$ would have been equivalent to 0.59 × [HO$_2$] and 0.44 × [HO$_2$], respectively. The RO$_2$ interference for the TMB experiments is likely to be due to a tiny fraction of the precursors, oxidation products and ozone passing through the denuders. Due to working at ppm concentrations the amount making it through the denuders could be enough to lead to a RO$_2$ signal. However, for the α-pinene experiments it seems that the denuders were more efficient at removing these precursors and oxidation products although the reasons for this remain unclear.

The following discussion was added to the text:

*"Although the denuders are efficient at removing gas phase species (Arens et al., 2001), it can be hypothesised that the signal was due to the formation of HO$_2$ and RO$_2$ radicals due to a small fraction of ozone, precursors and oxidation products passing through the denuders for the TMB experiments."*

*"The lack of interference for the α-pinene experiments suggests that the denuders were more efficient at removing the gas phase precursors and oxidation products from the chamber and that negligible concentrations of RO$_2$ species were present in the flow tube."*

Line 219: Discussion of correction for wall loss and non-plug flow would benefit from indicating directly the level of correction typically required beyond what can be inferred from Figure 3.

The average correction due to wall loss and non-plug flow was 22 %.

This has been clarified within the manuscript by adding the following text.

*"The average correction was 22%."*

Line 238: "which is only slightly larger than the diameter of HO2 (0.4 nm)." – is there any significance to this?

It is important to show that the model resolves the diffusion gradient of HO$_2$ despite its very small length scale. It was significant that the HO$_2$ only had to travel approximately the distance of its own diameter to go from being an adsorbed radical on the surface of the aerosol particle to a dissolved aqueous radical. It also only had to travel the distance of approximately its own diameter to move between bulk layers. This is really important when the reactions within the aerosol particles are extremely fast leading to large concentration gradients within the particle.

This has been clarified within the manuscript by adding the following text to the manuscript.

*"The bulk layer number was set to 100 corresponding to a bulk layer thickness of 0.5 nm which is only slightly larger than the diameter of HO$_2$ (0.4 nm) and implies that HO$_2$ only needs to travel approximately the distance of its own diameter to go from being an adsorbed radical on the surface of the aerosol particle to a dissolved aqueous radical. The same short distance must be overcome by HO$_2$ to move between bulk layers, which is important for convergence of the numerical model, especially when the chemical reactions within the aerosol particles are very fast compared to the diffusion time scales, leading to steep concentration gradients within the particle."*

**References**

Arens, F., Gutzwiller, L., Baltensperger, U., Gäggeler, H.W. and Ammann, M. (2001) Heterogeneous reaction of NO$_2$ on diesel soot particles, *Environ. Sci. Technol.*, **35**, 2191-2199.

Bedjanian, Y., Romanias, M.N. and El Zein, A. (2013) Uptake of $HO_2$ radicals on Arizona Test Dust, *Atmos. Chem. Phys.* , 6461-6471.

Behr, P., Scharfenort, U., Ataya, K. and Zellner, R. (2009) Dynamics and mass accommodation of HCl molecules on sulfuric acid–water surfaces, *Phys. Chem. Chem. Phys.*, **11**, 8048-8055.

Berkemeier, T., Huisman, A.J., Ammann, M., Shiraiwa, M., Koop, T. and Pöschl, U. (2013) Kinetic regimes and limiting cases of gas uptake and heterogeneous reactions in atmospheric aerosols and clouds: a general classification scheme, *Atmos. Chem. Phys.*, **13**, 6663-6686.

Berkemeier, T., Steimer, S.S., Krieger, U.K., Peter, T., Pöschl, U., Ammann, M. and Shiraiwa, M. (2016) Ozone uptake on glassy, semi-solid and liquid organic matter and the role of reactive oxygen intermediates in atmospheric aerosol chemistry, *Phys. Chem. Chem. Phys.*, **18**, 12662-12674.

Bones, D.L., Reid, J.P., Lienhard, D.M. and Krieger, U.K. (2012) Comparing the mechanism of water condensation and evaporation in glassy aerosol, *P. Natl. Acad. Sci.*, **109**, 11613-11618.

Davidovits, P., Kolb, C.E., Williams, L.R., Jayne, J.T. and Worsnop, D.R. (2006) Mass accommodation and chemical reactions at gas-liquid interfaces, *Chem. Rev.*, **106**, 1323-1354.

Davies, J.F. and Wilson, K.R. (2015) Nanoscale interfacial gradients formed by the reactive uptake of OH radicals onto viscous aerosol surfaces, *Chem. Sci.*, **6**, 7020-7027.

George, I.J., Matthews, P.S.J., Whalley, L.K., Brooks, B., Goddard, A., Baeza-Romero, M.T. and Heard, D.E. (2013) Measurements of uptake coefficients for heterogeneous loss of $HO_2$ onto submicron inorganic salt aerosols, *Phys. Chem. Chem. Phys.*, **15**, 12829-12845.

Gržinić, G., Bartels-Rausch, T., Berkemeier, T., Türler, A. and Ammann, M. (2015) Viscosity controls humidity dependence of $N_2O_5$ uptake to citric acid aerosol, *Atmos. Chem. Phys.*, **15**, 13615-13625.

Hanson, D.R., Ravishankara, A. and Solomon, S. (1994) Heterogeneous reactions in sulfuric acid aerosols: A framework for model calculations, *J. Geophys. Res. - Atmos.*, **99**, 3615-3629.

Hearn, J.D., Lovett, A.J. and Smith, G.D. (2005) Ozonolysis of oleic acid particles: evidence for a surface reaction and secondary reactions involving Criegee intermediates, *Phys. Chem. Chem. Phys.*, **7**, 501-511.

Houle, F., Hinsberg, W. and Wilson, K. (2015) Oxidation of a model alkane aerosol by OH radical: the emergent nature of reactive uptake, *Phys. Chem. Chem. Phys.*, **17**, 4412-4423.

Lakey, P.S.J., George, I.J., Whalley, L.K., Baeza-Romero, M.T. and Heard, D.E. (2015) Measurements of the $HO_2$ Uptake Coefficients onto Single Component Organic Aerosols, *Environ. Sci. Tech.*

Marshall, F.H., Miles, R.E., Song, Y.-C., Ohm, P.B., Power, R.M., Reid, J.P. and Dutcher, C.S. (2016) Diffusion and reactivity in ultraviscous aerosol and the correlation with particle viscosity, *Chem. Sci.*

Matthews, P.S.J., Baeza-Romero, M.T., Whalley, L.K. and Heard, D.E. (2014) Uptake of $HO_2$ radicals onto Arizona test dust particles using an aerosol flow tube, *Atmos. Chem. Phys.*, **14**, 7397-7408.

Pfrang, C., Shiraiwa, M. and Pöschl, U. (2010) Coupling aerosol surface and bulk chemistry with a kinetic double layer model (K2-SUB): oxidation of oleic acid by ozone, *Atmos. Chem. Phys.*, **10**, 4537-4557.

Pöschl, U., Rudich, Y. and Ammann, M. (2007) Kinetic model framework for aerosol and cloud surface chemistry and gas-particle interactions - Part 1: General equations, parameters, and terminology, *Atmos. Chem. Phys.*, **7**, 5989-6023.

Power, R., Simpson, S., Reid, J. and Hudson, A. (2013) The transition from liquid to solid-like behaviour in ultrahigh viscosity aerosol particles, *Chem. Sci.*, **4**, 2597-2604.

Price, H.C., Murray, B.J., Mattsson, J., O'sullivan, D., Wilson, T.W., Baustian, K.J. and Benning, L.G. (2014) Quantifying water diffusion in high-viscosity and glassy aqueous solutions using a Raman isotope tracer method, *Atmos. Chem. Phys.* , **14**, 3817-3830.

Schwartz, S. and Freiberg, J. (1981) Mass-transport limitation to the rate of reaction of gases in liquid droplets: Application to oxidation of $SO_2$ in aqueous solutions, *Atmos. Environ.*, **15**, 1129-1144.

Shiraiwa, M., Pfrang, C. and Pöschl, U. (2010) Kinetic multi-layer model of aerosol surface and bulk chemistry (KM-SUB): the influence of interfacial transport and bulk diffusion on the oxidation of oleic acid by ozone, *Atmos. Chem. Phys.*, **10**, 3673-3691.

Shiraiwa, M., Sosedova, Y., Rouvière, A., Yang, H., Zhang, Y., Abbatt, J.P., Ammann, M. and Pöschl, U. (2011) The role of long-lived reactive oxygen intermediates in the reaction of ozone with aerosol particles, *Nature chemistry*, **3**, 291-295.

Slade, J.H. and Knopf, D.A. (2014) Multiphase OH oxidation kinetics of organic aerosol: The role of particle phase state and relative humidity, *Geophys. Res. Lett.*, **41**, 5297-5306.

Steimer, S.S., Berkemeier, T., Gilgen, A., Krieger, U.K., Peter, T., Shiraiwa, M. and Ammann, M. (2015a) Shikimic acid ozonolysis kinetics of the transition from liquid aqueous solution to highly viscous glass, *Phys. Chem. Chem. Phys.*, **17**, 31101-31109.

Steimer, S.S., Berkemeier, T., Gilgen, A., Krieger, U.K., Peter, T., Shiraiwa, M. and Ammann, M. (2015b) Shikimic acid ozonolysis kinetics of the transition from liquid aqueous solution to highly viscous glass, *Phys. Chem. Chem. Phys.*

Vieceli, J., Roeselova, M., Potter, N., Dang, L.X., Garrett, B.C. and Tobias, D.J. (2005) Molecular dynamics simulations of atmospheric oxidants at the air-water interface: Solvation and accommodation of OH and $O_3$, *J. Phys. Chem. B*, **109**, 15876-15892.

Zobrist, B., Soonsin, V., Luo, B.P., Krieger, U.K., Marcolli, C., Peter, T. and Koop, T. (2011) Ultra-slow water diffusion in aqueous sucrose glasses, *Phys. Chem. Chem. Phys.*, **13**, 3514-3526.